# Holistic Sexual-Reproductive Healthcare Services and Needs for Queer Individuals: Healthcare Providers’ Perspectives

**DOI:** 10.3390/healthcare12101026

**Published:** 2024-05-15

**Authors:** Raikane James Seretlo, Hanlie Smuts, Mathildah Mpata Mokgatle

**Affiliations:** 1Department of Public Health, Sefako Makgatho Health Sciences University, Pretoria 0208, South Africa; mathildah.mokgatle@smu.ac.za; 2Department of Informatics, University of Pretoria, P.O. Box X323, Pretoria 0007, South Africa; hanlie.smuts@up.ac.za

**Keywords:** LGBTQIA+, queer, sexual-reproductive healthcare, services and needs, healthcare providers, perspectives

## Abstract

There are ongoing debates and controversies about whether genderqueer individuals have specific sexual-reproductive healthcare services and needs (SRHSNs). This study intended to identify and explore queer-specific SRHSNs among healthcare providers (HCPs) in Gauteng Province, South Africa. This was an exploratory sequential mixed-methods study, and this article focuses on the qualitative findings of that investigation. Thirty-three HCPs were purposively sampled, and semi-structured one-on-one interviews were used to collect data between September and November 2023. The data were analyzed using thematic content analysis (TCA). The results of this study revealed nine main themes: a crucial need for inclusive healthcare facilities; a need for psychological, counseling, and therapeutic support in sexual and reproductive healthcare; access to sexual-reproductive education and integrating support; suggested reproductive health services for queer sexual wellness; improved accessibility and particular queer reproductive healthcare; optimizing services related to human immunodeficiency virus (HIV), pre-exposure prophylaxis (PrEP) access, and sexually transmitted illness (STI) treatment; genderqueer persons’ parenthood aspirations and empowerment; the safe availability of intimacy tools; and navigation transitions. A holistic and inclusive healthcare approach that fits psychological support, comprehensive sexual-reproductive education, and specialized services to accommodate the unique needs of queer individuals should be implemented and made easily accessible.

## 1. Introduction

Recognizing sexual-reproductive healthcare services and needs toward attaining holistic well-being for every individual is paramount in fostering a society where everyone can thrive. It is imperative to recognize and address the uniqueness of genderqueer individuals’ sexual-reproductive healthcare services and needs (SRHSNs) not only as a matter of inclusivity but also as a fundamental step toward achieving true equality and well-being for all. Queer is an umbrella term for inclusive expression; it refers to anyone who feels excluded because of their sexual practices or who rejects the heteronormative sex/gender/sexual identification system [1]. Lesbians, gay people, bisexual people, transgender people, men who have sex with men (MSM), and intersex people are all examples of genderqueer individuals [2].

Several research scholars outline that genderqueer individuals may require and face unique sexual-reproductive healthcare services, needs, and challenges [2,3,4]. All people, including those who identify as queer-fluid, need sexual and reproductive healthcare [5]. Furthermore, SRHSNs, such as HIV and STI testing, cervical cancer screening, and contraceptive care, are critical for the prevention, identification, and treatment of a variety of sexual and reproductive health consequences [6]. In addition, the provision and inclusion of other services and needs for genderqueer individuals would improve their health and well-being by preventing unintended pregnancies [7], lowering the increased risk of unplanned pregnancies and abortions [8], and mitigating high-risk sexual behaviors such as having several sexual partners, switching partners, irregular condom use, unprotected anal sex, and transactional sex [9]. Providing these services coincides with multiple sustainable development goals (SDGs) for 2030, including SDG 3, Good Health and Well-Being; SDG 5, Gender Equality; and SDG 10, Reduce Inequalities [10].

However, some healthcare providers (HPCs) do not regard SRHSNs for genderqueer individuals as specific and unique [11], which means certain HCPs are unable to understand and respect genderqueer individuals’ specific and unique healthcare needs fully, particularly regarding SRHSNs. According to some research, some HCPs’ heteronormativity and cisnormativity approaches continue to be barriers to gender-fluid individuals accessing SRHSNs [12]. Treatment guidelines are also considered heteronormative [13]. For example, HCPs continue to perceive and assume genderqueer individuals incorrectly as heterosexual [14,15], and HCPs regard most genderqueer individuals to be the same as any other patient [11], even though some gender-fluid individuals indeed have needs like gender transition, which includes gender-affirming surgeries and hormone therapy [16].

Other barriers to specific SRHSN access for genderqueer individuals include HCPs lacking healthcare competence concerning reproductive health priorities and treatment [16], and a scarcity of education and training for HCPs on genderqueer health-related problems [17], such as gender-affirming healthcare services like hormone therapy and gender-affirming surgeries and a lack of tailored knowledge to assist genderqueer individuals [12]. Furthermore, some barriers that inhibit genderqueer individuals from accessing their specific SRHSNs include HPCs using discriminatory words and treatment, displaying prejudice, and stigmatizing such individuals [12,16,17,18].

If HCPs are unfamiliar with, lack understanding of, and are not knowledgeable about SRHSNs for genderqueer individuals, it increases risks for genderqueer individuals. Such risks entail the following: the prevalence of STIs (especially HIV and acquired immunodeficiency syndrome [AIDS]) would rise and remain high [18,19]; genderqueer individuals might have sexual issues such as lack of access to fertility preservation and gender-affirming hormonal therapy, and unintended pregnancies; and they might suffer from sexual harassment [18], indulge in risky sexual practices such as having numerous sexual partners and sharing partners, use condoms inconsistently, engage in unprotected anal sex, pursue transactional sex [9,20], and practice high-risk sexual behaviors like performing or receiving “rimming” and exchanging sexual devices [20]. In addition, HCPs’ discriminatory attitudes toward genderqueer individuals might prevent them from accessing healthcare services [21], leading to uncertainty about whether or not they would be approved for assisted reproduction [15], placing some genderqueer individuals at a higher risk of unintended pregnancies [7]. Little is known about the specific and unique SRHSNs for genderqueer individuals. Hence, this research intended to identify and explore queer-specific SRHSNs among HCPs in Gauteng Province, South Africa. This research further aimed to enhance genderqueer individuals’ access to SRHSNs and enable HCPs to serve genderqueer individuals with their specific and unique SRHSNs better, thereby improving and promoting their overall well-being.

## 2. Materials and Methods

In this section, we outline the procedures and analytical techniques employed to investigate the research question (Figure 1).

### 2.1. Study Design

This was an exploratory sequential mixed-methods study, and this article focuses on the qualitative findings of the investigation. Exploratory qualitative research is undertaken when little is known about a phenomenon or a topic that has not been fully defined [22]. This paper presents the qualitative findings on identifying and exploring queer-specific SRHSNs among HCPs in Gauteng Province, South Africa, to increase genderqueer individuals’ access to SRHSNs.

### 2.2. Study Context

This study was conducted in the Gauteng Province of South Africa (SA) at seven district hospitals. These settings were selected because all the district hospitals in SA render secondary levels of care, such as family medicine, pediatrics, obstetrics and gynecology, emergency, and outpatients. These district hospitals receive outreach as well as backing from general specialists stationed in regional healthcare facilities, implying that they are typically smaller institutions that receive support and help from general specialists working in larger regional hospitals. After receiving ethical clearance from the Sefako Makgatho Health Sciences University Research Ethics Committee (SMUREC) in September 2023, the principal investigator requested permission to conduct this study at the National Health Research Database (NHRD), which was forwarded to the hospital’s chief executive officers (CEOs).

### 2.3. Population, Sampling, and Recruitment

The participants in this study were 33 HCPs from seven selected district hospitals in Gauteng Province, South Africa. The specific represented participants (HCPs) who agreed to take part in this study included medical doctors, nurses, psychologists, and social workers. The reason for selecting these participants was because they were closely dealing with the SRHSNs of the vast number of patients in healthcare facilities, including genderqueer individuals (refer to Table 1). This study employed a purposive sampling method to select the participants who had agreed to take part in our study [23]. We chose this method because HCPs have experiences in caring for different types of patients including queer individuals. Therefore, they were the relevant population to be purposively sampled due to their expertise, and we anticipated them providing rich and detailed insights regarding the specific SRHSNs of queer individuals. We used a systematic approach to choose and select HCPs who rendered SRH to different patients in the healthcare facilities as we anticipated that they might be having experiences and encountered the provision of SRH to queer individuals too. The study population was guided by data saturation, which means that the principal investigator stopped collecting data once no new information was coming up from the participants. Data saturation was reached at the 30th interview, but the principal investigator continued with three more additional interviews to confirm that data saturation had indeed been reached.

The recruitment of participants occurred in three different ways during working hours based on the setup of each study setting. First, the principal investigator met the participants during their prayer time in the morning, where they shared the purpose and objective of this study with all staff members. Those who agreed to participate at that time confirmed appointment dates and times for the data collection. Second, the principal investigator confirmed appointments with the participants one day before data collection. Lastly, the overall purpose of this study was presented at staff meetings, and more participants were recruited by visiting wards and consulting rooms. All who had agreed to participate in this study gave verbal and written consent.

### 2.4. Inclusion and Exclusion Criteria

The initial planned inclusion criteria were to include all HCPs working at the selected seven district hospitals in Gauteng Province. However, one study setting did not provide permission to conduct this study, and, as a result, this study included only HCPs in the seven district hospitals that had indeed provided permission. HCPs who were not working at the seven selected district hospitals were excluded from this study.

### 2.5. Data Collection

Data were gathered through semi-structured, one-on-one interviews from September to November 2023. The sessions were facilitated by a semi-structured interview guide and recorded using a digital recorder. The semi-structured interview guide had two main sections: demographic and in-depth, open-ended questions. English was the language used to collect data; however, participants were allowed to respond in their home languages, although all 33 participants mostly responded in English except when linking ideas (see the quotations in Section 3, Results).

A pilot study was conducted in September 2023 at one of the seven study settings and comprised three participants. The findings emanating from the pilot study were excluded from the main study as its purpose was to refine the desired methods, instruments, and procedures. The pilot study was led by the principal investigator and supported by supervisors as peer reviewers to check whether the interview guide answered the anticipated research question. The interview guide was amended and confirmed by all authors. A few modifications were made to the questions or wording based on responses from the HCPs in the pilot study, but no questions were deleted.

The main data collection of this study commenced in September 2023 and continued during September, October, and November 2023. The researcher obtained informed consent from the participants before data collection and began with demographic data written in the consent form, not recorded on the digital recorder, but the names of the participants were excluded to protect their confidentiality and privacy.

The semi-structured one-on-one interviews were held in a separate room at the institution, allowing HCPs to engage in the sessions without interruption. Each interview lasted between 20 and 50 min. The main question was “Please tell me about the time when queer individuals come to the hospital for sexual-reproductive healthcare services and needs”. The principal investigator used probing questions to collect more in-depth data, e.g., “What are the types of services that you will provide to them?”, “What are the specific sexual-reproductive healthcare services and needs for queer individuals?”, and “What other specific sexual-reproductive healthcare services and needs do you consider important for queer individuals?”. The researcher recorded any nonverbal cues for data analysis usage, such as smiles, mannerisms, and voices in the field notes.

This study was conducted with one principal investigator and the research assistant, who holds a master’s degree in public health. The researcher received refresher training on how to conduct in-depth interviews using an interview guide for data collection from the main supervisor during the master’s studies and prior to the data collection of this study. The principal investigator acquires knowledge on qualitative studies through students’ supervision, which enhanced in-depth interactions with participants. The research assistant holds honor’s degrees in industrial psychology and was trained by the principal investigator on data collection steps and on the conduction of interviews.

### 2.6. Data Analysis

The collected data were analyzed through thematic content analysis (TCA), following the four steps by Ravindran [24]. First, the principal investigator prepared data together with the transcriber, and audio data were transcribed using the Philips SpeechExec Pro Version 10 transcription tool, some of which were later translated, corrected, and edited. All 33 recorded interview sessions were transcribed verbatim, and the demographic data were entered into an MS Excel sheet (Microsoft (365)). Second, the principal investigator repeatedly read the organized transcripts, thus immersing themselves in the data. Third, the principal investigator used NVivo V12 software to code the data following two steps: The first coding step was used to classify and categorize the different codes and the principal investigator shared the first coding with one supervisor who acted as a peer reviewer to confirm the creation of codes. Additionally, the principal investigator developed a manual codebook which was later imported into NVivo V12. NVivo 12 is a piece of software used for qualitative data, and it enables the researchers to import transcripts, create themes, perform coding, and collaborate between students and supervisors. The second coding step commenced with the clarification and grouping of similar codes to develop themes, which was first carried out by the principal investigator and confirmed by supervisors. Finally, themes were developed, whereby the researcher grouped codes from the second coding with similar meanings and interpreted them into full meaningful phrases, thus making them more concise. Conclusions were drawn from the direct quotations from participants without changing their meanings.

### 2.7. Trustworthiness

To ensure the trustworthiness of this study, we strived to enhance its transferability, credibility, confirmability, and dependability, as proposed by Korstjens and Moser [25]. For transferability, the principal investigator collected data from seven different district hospitals in the Gauteng Province of SA to generalize the study findings and, thus, be representative of Gauteng provincial hospitals. In addition, the principal investigator provided a detailed research process for other scholars to use in research projects. Credibility was achieved by data collection training for one research assistant and collaborating during data collection to ensure rich, high-quality data. Further, a pilot study of three interviews was conducted using the interview guide before the main study commenced. Conformability was achieved and maintained, the researcher secured the signed consent forms, and the data analysis process was followed and recorded step by step. Moreover, from the beginning to the end of this study, the principal investigator meticulously noted in a notebook any steps taken toward their mitigation limitations and biases, ensuring that researcher and research assistant biases were removed and the HCPs’ perspectives on genderqueer individuals’ SRHSNs were accurately conveyed. Lastly, the principal investigator ensured dependability by acting as a peer reviewer for the research assistant daily after data collection and included both supervisors for reviewing the collected data, analyzing data (like transcripts), and using instruments such as a reliable digital recorder. One supervisor acted as an adjunct coder to confirm the codes and themes.

### 2.8. Ethical Considerations

This study’s research proposal was first presented at the Department of Public Health Research Committee (DPHRC), whereafter it was submitted and presented to the School of Healthcare Sciences Research Committee (SREC). The research proposal subsequently received approval and ethical clearance to conduct this study from the SMUREC, with protocol number SMUREC/H/291/2023: PG. Upon receiving ethical clearance, the researcher applied for permission to conduct this study at the selected Gauteng District Hospitals via the NHRD website.

Participants who had agreed to participate in this study were supplied with informed consent forms and comprehensive study details, such as its aims and objectives. They were also informed that their participation was entirely optional and that they might withdraw at any time throughout the interview without consequence. The names and surnames of the participants were not included in the interview guide, field notes, or recordings; instead, the participants were assigned numbers for identification (e.g., P1, representing Participant 1), and all data tools were strictly secured to ensure safety, privacy, and confidentiality.

To comply with the Protection of Personal Information Act (POPIA), the principal investigator acquired informed consent from the participants, collected limited personal information, and securely stored and regulated access to the data, which would be retained for five years. Lastly, the principal investigator protected the participants’ personal information by not disclosing their information to third parties, and this study’s findings were only shared with supervisors and transcribers.

## 3. Results

In this section, we present the results of this research study. These include the demographic data, themes, and sub-themes (Table 2).

Our study included 33 HCPs as participants, with five identifying as male and 28 as female. The participant’s mean age was 36.7. When questioned about their marital status, 22 were single and 11 were married. All of them had tertiary education and were HCPs; and there were five social workers, six clinical psychologists, seventeen nursing staff with different categories, and four medical doctors. The longest duration of work was 36 years, the middle duration of work was 25 years, and lowest duration of work was 1 year.

### 3.1. The Crucial Need for Inclusive Healthcare Facilities

Most participants recognized the importance of the inclusivity of healthcare facilities for genderqueer individuals. Furthermore, the participants suggested a need for gender-neutral bathrooms and the importance of genderqueer-friendly services.

#### Genderqueer-Inclusive Bathrooms

The HCPs raised concerns about inadequate accommodations for genderqueer individuals in government healthcare facilities, notably in male and female bathrooms. They noted the issues faced by these individuals, such as discomfort when using bathrooms in healthcare institutions, particularly those identifying as lesbians who were uncertain about which facility to use and frequently inquired about it. They proposed the establishment of bathrooms specifically for genderqueer people within government hospitals, as well as the provision of clear indications and guidance for gender-fluid people to use facilities for disabled persons that do not bear gender-specific labels, thus providing a more inclusive and comfortable space that aligns with their gender identity. In addition, the participants underlined the critical need for more inclusive facilities that better meet the specific needs of genderqueer individuals in healthcare settings.

“*Our government hospitals do not cater them well (sic), the reason I am saying this is, if you look [at] our toilets are labelled male and females. Let’s say there is lesbian patient coming to the hospital, feels I’m a man, where does she supposed to go now; there’s a toilet labelled male, there’s a toilet labelled female. Ya neh!! in most cases some queer patients will come to us and ask, “Which toilet must I use?” and indeed, which one must they us—so whenever they come to me, I always show them disabled toilets because disabled—it doesn’t choose which one is male, which one is female, they just go*”.(Participant 22, enrolled nurse)

“*You know sometimes I ask myself, when somebody lives like a queer, neh, and the next thing he is seen as a male by everybody and expected to go where males go, I am going to give you example, in terms of the bathrooms, public bathrooms or something like that, they would go to their same-sex bathrooms, but the question would be, are they comfortable? Why can’t we do what malls does whereby there are family bathrooms? Government should do something like that for queer have special bathrooms for them so that it’s clearly indicated—I don’t know but I am not sure if you get what I mean, because I don’t know if they feel comfortable in using the normal ones that we normally go but, yah*”.(Participant 26, clinical psychologist)

### 3.2. The Need for Psychological, Counselling, and Therapeutic Support in Sexual and Reproductive Healthcare

The HCPs agreed on the crucial need to incorporate psychological, counseling, and therapeutic care for genderqueer individuals. The majority of the HCPs emphasized the importance of thorough counseling, particularly for persons suffering physical changes as part of their gender transition. The HCPs also emphasized the importance of assessing readiness and providing sexual change counseling, delivering information on both the benefits and disadvantages. Some of the HCPs underlined the critical role of psychologists in providing psychological counseling to those navigating gender changes. Recognizing their psychological composition and emphasizing the significance of knowing the patient’s coping methods demonstrate a holistic approach to gender-affirming healthcare.

The HCPs further provided information on the psychological aspects of many healthcare scenarios. Recognizing the potential psychological consequences of such experiences, they explored the referral of lesbian clients for psychological therapy after terminating a pregnancy. The HCPs acknowledged that genderqueer individuals were a vulnerable population and experienced many sexual challenges, such as rape, which might require the termination of a pregnancy, especially for lesbian clients who were not ready to have children. In addition, the HCPs considered psychological counseling related to sexual-reproductive health as important as they indicated that, at times, they provided it prior to antiretroviral therapy commencement. Lastly, the HCPs addressed the mental health difficulties gay people might encounter, such as substance abuse and mental breakdown caused by societal rejection due to their sexual orientation. Most HCPs emphasized the importance of counseling and educational lectures, suggesting a shared understanding among HCPs of the complex interplay between sexual needs and mental well-being in the context of gender-fluid healthcare.

*“I think they need counseling, yes, we do offer counseling to them and a lot of counseling. We do it when queer people want to change themselves as persons, image change and everything you know, and some people just want to see themselves there, having breasts even if it’s a man, or having a structure of a woman. So, we assess and see how ready this person is that she is talking about, we provide them with sexual change counseling by giving them information on the advantages and disadvantages*”.(Participant 13, social worker)

*“I think psychologists, yeah psychological counseling. They will need to see a psychologist because imagine now this patient has changed from male to female, she is still adapting though it’s something that he or she wanted, but then you need to give a psychological, get a psychological breakdown and how is he or she is taking it now*”.(Participant 18, registered nurse)

*“So, we tend to provide a lot of educational talk and counselling because you have others who move from just a normal life and eventually, they are substance users due to their sexuality challenges. Eventually they tend to have some sort of mental breakdown because of that ill-acceptance, not only from the community but the families that they are in as well their sisters, brothers, mothers, fathers, yah*”.(Participant 20, Social worker)

### 3.3. Access to Sexual-Reproductive Education and Integrating Support

Most HCPs mentioned that they provided education on sexual-reproductive education as a crucial holistic SRHSN for genderqueer individuals. They further believed that integrated support was essential to protecting and ensuring inclusivity for genderqueer individuals. The HCPs further outlined types of sexual education, such as information on prophylaxis for sexually transmitted infections (STIs). They stressed that regardless of the provision of sexual and reproductive education, genderqueer individuals should practice safe sexual activities and adopt protective measures; yet, the HCPs continue educating such individuals on responsible behaviors, such as avoiding risky situations with strangers, seeking medical help as soon as possible after sexual assault, and understanding the risks of engaging in sexual activities without protection.

Some HCPs highlighted contraceptive education and the importance of using protection like condoms, not only to prevent pregnancies but also the transmission of diseases as SRHSNs for genderqueer individuals. The HCPs mentioned that they provided education on practicing safe sex, such as involvement in discussions about the use of condoms and other preventive measures.

*“They need access to education, it’s more education with regard to prevention of our burden of diseases, like any other person, and we always teach them about abstinence, condoms, and all that*”.(Participant 4, social worker)

*“I don’t think we should build another or a different clinic for them, but what do you call it? An image or pamphlet. Yes, if we can have a pamphlet is posted here so that they feel welcome as well, they will then be educated and informed on how things end up in resulting STIs, infections and know males do not like talking a lot. Nonetheless, we are giving them health education on HIV prophylaxis medication, and we normally teach them how to protect themselves like don’t go home late, don’t stay with strangers because people will be taking advantage, and then we give them education that if they get raped, they should come to the hospital within 72 h, because after 72 h we can’t give you any treatment like those I said prophylaxis for the STIs*”.(Participant 12, registered nurse)

*“Practicing safe sex will be one of the educations that I give the talks, practicing safe sex and, yeah. It is important*”.(Participant 32, medical doctor)

### 3.4. Holistic Approaches to Genderqueer Persons’ Sexual Wellness

Most HCPs highlighted that the sexual and reproductive well-being of genderqueer persons must be addressed comprehensively. The participants believed that regardless of their sexual practices and uniqueness, some important examinations and services cannot be neglected, as a gender-fluid person’s biological sex remains the same as that of a heterosexual person. HCPs emphasized that genderqueer individuals deserve to be treated holistically in the provision of SRHSNs.

#### 3.4.1. Breast Examinations

Many HCPs mentioned that lesbian women should be provided with services such as contraceptives that are related to women’s well-being regardless of their sexual orientation. Further, the participants emphasized that breast examination, health education on how to examine their breasts, and mammograms are some of the important SRHSNs for lesbian women.

“*The only other services that I can offer lesbian women is if they maybe have any illnesses, you know, because they are women; they do have breasts, so they might have problems with their breasts, they might have problems with their menstrual cycle, so all those woman things, you know, woman illnesses, so those things we can do. We can do mammogram for them*”.(Participant 9, registered nurse)

“*They also need breast examination [,] I would provide breast examination*”.(Participant 11, registered nurse)

“*I think we also need to do, what do you call this, a cancer screening basically like your breast cancer and teach them how to do the mammogram because akere they are lesbians, they have breasts, they could be prone, especially if they are smoking. Yes, mammogram, information on how to examine their breast just to exclude the breast cancer*”.(Participant 18, Registered nurse)

#### 3.4.2. Male Circumcision

The HCPs affirmed that male circumcision for gay men was one of their SRHSNs. They indicated that circumcision is important as most gay men present with STIs at the healthcare facilities and this procedure would minimize the risk of infection. However, some HCPs questioned whether or not gay men had genitals. Nevertheless, they mentioned that if gay men had an actual penis, they would provide circumcision at the district hospital and if not, they would be referred for advanced care.

“*I could say male circumcision, because most of our patients come with STIs, whereby they will say where can we get the services for male circumcision? I think male circumcision might be the one*”.(Participant 6, registered nurse)

“*Oh, year for gay men, circumcision. Yes!! They also need a male medical circumcision as one of their sexual health*”.(Participant 11, Registered nurse)

“*Eish, usually what is happening like, do they have an actual penis or do they get a penis from someone else, or how does it work with regard to them because I’m thinking if someone comes here for me, firstly before I can offer the services of circumcision I have to assess the penis if it’s something that we can do at a district level, otherwise anything else we just refer to the higher institutions where there are specialists*”.(Participant 16, registered nurse)

#### 3.4.3. Pap Smears

The HCPs noted pap-smears as one of the vital SRHSNs for genderqueer individuals and focused mainly on transgender men and lesbians. The HCPs acknowledged that transgender and lesbian individuals should be examined and excluded from diseases such as cancer and indicated that conducting pap-smear examinations would help them in preparation for fertility.

“*I would provide pap-smear, yes pap-smear, if the client is transgender*”.(Participant 11, registered nurse)

“*So, well you don’t have to say because a patient was a male, now let’s the patient is transgender, they do have woman private parts so they deserve all the tests such the ones of checking cancer of the cervix, yes they should also through pap-smear, you, see?*”.(Participant 15, registered nurse)

“*Pap smear, akere because we also want to protect them from cervical cancers and then the other thing that we can do, if-akere they want kids, akere, then they can get pregnant in any other ways anyway*”.(Participant 24, Registered nurse)

#### 3.4.4. Prostate Services

Most of the HCPs stipulated that genderqueer individuals should continue visiting healthcare facilities for check-ups and receive prostate services to exclude infections such as urinary tract infections (UTIs), challenges with genitals, and prostate cancers.

“*A gay patient, for sexual and reproductive… because he is a man, right? So, they’ve got prostate, at times they have got urine tract infections, they should be examined*”.(Participant 9, registered nurse)

“*We want to see your genitals, if you are having problems with your genitals, are you having problems with your inside genitals, then from outside inside we want to see things like your prostate, you know, you do a rectal exam*”.(Participant 17, medical doctor)

#### 3.4.5. Urological and Vasectomy Services

The majority of the HCPs suggested urological procedures and the removal of the vas deferens as one of the crucial SRHSNs for queer individuals.

“*Yoh! Yah, most of them they will want the vasectomy especially the male ones. I think vasectomy is the service that they might be given because, yah, as partners when it comes to sexually you need both of you to be involved in it*”.(Participant 6, registered nurse)

“*I think some of queer clients would even want to get sterilization. Yes, they will want their male parts to be sterilized*”.(Participant 8, Registered nurse)

“*Gays, I really don’t know, unless it will be for the vasectomy part, unless they specify their- what can I say? The reason for coming on that day, then we will see from there*”.(Participant 27, registered nurse)

### 3.5. Improved Accessibility and Particular Queer Reproductive Healthcare

Most of the HCPs outlined that comprehensive queer reproductive healthcare services, needs, and resources must be enhanced. In addition, the HCPs stipulated that genderqueer individuals deserve services, such as the choice to terminate a pregnancy, being informed about condoms and being able to access them effortlessly, and family planning, including different contraceptive methods, which are some of such SRHSNs.

#### 3.5.1. Choice of Termination of Pregnancy

Many participants pointed out that genderqueer women have the right to choose to terminate their pregnancy (CTOP) as one of their SRHSNs. Furthermore, most participants emphasized that this service is a need and must be rendered since genderqueer women tend to experience various social challenges, such as unplanned pregnancy, rape, abuse, and assault. Again, the HCPs stated that genderqueer women could be at risk of falling pregnant and might be fearful of informing their partners and highlighted the importance of making this service available since some lesbians fear being judged.

“*OK. Yeah. Especially the lesbians. Yes, lesbians are females. Yes, they do have their reproductive system. In our society they know her as lesbian, sometimes because of the cruelty of this world, they will forcefully take her to publish (sic) her, they rape her, and she becomes pregnant. She also needs that CTOP services because it affects her because according to her, she prefers to be a lesbian not to sleep with men and have a baby or become a mother*”.(Participant 3, registered nurse)

“*Yes, so I’ve had two different cases, the other one was a victim of sexual assault, and then the other one I think maybe because she did tell me that she has been lesbian for her whole and then she just sex with a male for the first time and she fell pregnant, so she is actually afraid of the partner because it’s another female, what will she say because she knows that they are both lesbian, there is no way one can impregnate the other; so that was the reason why she wanted termination*”.(Participant 8, registered nurse)

“*Bear in mind anything can happen anywhere. You can get- being raped and now you are feeling like you are- I’m a shemale, I’m a male, I’ve got a female partner, now somebody raped me, then you fall pregnant. The best way, prevent; even that incident happen you know I am protected, I don’t think you will feel comfortable while feeling that they are male now they must go through pregnancy, and still, they must go through abortions while they feel that they are male*”.(Participant 22, Enrolled nurse)

#### 3.5.2. Contraceptive Accessibility and Promotion

The HCPs displayed a variety of viewpoints regarding the importance of promotion and education about the use and accessibility of condoms for all genderqueer individuals. The HCPs further emphasized the necessity for genderqueer individuals to practice safe sex and thus considered condom use and accessibility as an SRHSN for genderqueer individuals. In addition, the HCPs recognized other family planning approaches as important SRHSNs for genderqueer individuals. Notably, the HCPs indicated that contraceptives were also lesbian needs and stipulated oral contraceptive pills and implants as some of the contraceptives that could help lesbians stop and control their menstrual cycles, thus allowing them to feel comfortable with their sexual orientation.

“*We can only provide condoms for everyone yes, I think, even the gays we do provide condoms for them*”.(Participant 8, registered nurse)

“*They need is also to access condoms. Like yeah for safe sex to in order to have safe sex*”.(Participant 2, clinical psychologist)

“*The type of service that we give to gay men, we offer them condoms, that’s the only service that we can give them. We offer them condoms to protect themselves. Imagine a lesbian going on their dates monthly, I don’t think they would like to see their menses since they feel like man, implants, and the use a Depo-Provera, a Depo-Provera is an injectable for three months, but it will also help them to regulate their menses because most when you use them that you don’t go for menstruations. It’s good because I’m a shemale, I feel I’m a male, now I have- there is this time of the month where I must buy pads, but if you are using this there is no need to buy pads. […] are also important. That’s the service that we can offer for them*”.(Participant 22, enrolled nurse)

### 3.6. Optimizing Services Related to HIV, PrEP Access, and STI Treatment

Many HCPs asserted that sexually related infection services and the risks of acquiring prevention should be maximized for genderqueer individuals. Again, the HCPs stated that genderqueer individuals deserved access to HIV prevention, such as PrEP and PEP medicines, pre- and post-HIV testing counseling, and understanding STIs.

#### 3.6.1. HIV Preventative Measures and Prophylaxis Accessibility

Most HCPs identified essential medication that could prevent the spread of HIV throughout genderqueer people’s bodies and others that could be used when genderqueer individuals have possibly been exposed to HIV. The HCPs indicated that these treatments and medicines are essential as genderqueer individuals are adventurous. Lastly, the majority of HCPs linked pinpointed HIV services to important needs for genderqueer individuals, including pre-and post-counseling before HIV testing.

“*Queer individuals’ needs are them having accessibility to prophylaxis like I’ve mentioned in case there is sexual assault. This will help prevent illnesses such as HIV*”.(Participant 2, clinical psychologist)

“*HIV test why am I forgetting HIV testing? in fact for all of them HIV test. HIV test first and foremost. I think we should give PEP, we give PrEP, so that’s what I will do for a lesbian woman. Again, I think they need to have access to especially post exposure prophylaxis because they are promiscuous, and if they do those things at least can they rather be okay after doing those things because if we say PrEP then we are not really sure if they will go, or they will drink it for that time that they are supposed to and whatever*”.(Participant 11, Registered nurse)

“*Sexual reproductive health besides contraceptives? I don’t know if our PrEP and PEP, I think we can offer that because I mean they– yah, some patients are– queer people are adventurers, I can say, [Laughter] and they are proudly so, so PEPs and our PrEP we also offer that beside contraceptives. I think that’s one important thing that we also need to emphasize on HIV testing is part of that, because I’m currently working at the clinic with HIV and all that; we also assist with those*”.(Participant 32, medical doctor)

#### 3.6.2. STI Awareness and Treatment

Some HCPs noted that STI information, treatment, and assessment are essential healthcare needs for genderqueer individuals. Further, the HCPs indicated the benefits of STI services, such as protecting genderqueer individuals who might be raped from “corrective” rape or another justification.

“*I also think they need treatment for STIs and oh yes, treatment for STIs and screening. You know they are at risk of being raped by men who want to punish them for their sexual orientation and most perpetrators uses rape to try change queer individuals*”.(Participant 8, registered nurse)

“*We can screen them for STIs. So, if we render prevention for them, just for protection for the perpetrators out there who can give them STIs through rape. STI is curable but pregnancy you can’t cure it*”.(Participant 22, enrolled nurse)

### 3.7. Genderqueer Individuals’ Parenthood Aspirations and Empowerment

Most of the HCPs believed that genderqueer individuals deserved to have children of their own and become parents. This aspect was one of the noted sexual-reproductive needs, namely that regardless of such individuals’ sexual practices and sexual and gender activities, they had the right to conceive should they so wish. Accordingly, most of the HCPs identified different pathways through which genderqueer individuals could become parents to children of their own and that they should be encouraged to try various fertility options.

#### 3.7.1. Adoption Options

This study uncovered many HCPs’ varied perspectives on adoption for married, committed, or any genderqueer individuals who wished to have children. The HCPs indicated different avenues through which genderqueer individuals could adopt babies if they did not wish to bear children themselves.

“*I’ve got several people that I gave babies for adoption, a woman and a woman, lesbians. They are raising children, and they are raising them well like a normal couple, you understand? Because they approached me. Remember, for me, the best interests of the child come first. And the children that maybe I give out or I recommend out for people to raise, I don’t recommend people because they are lesbian, they are gay. I recommend people because they want to be parents*”.(Participant 13, social worker)

“*So, we tend to also provide advice on as much as you guys are engaged as female and female or whatever, and you guys are having a sexual relationship, it doesn’t mean you guys cannot have a family, you understand. We tend to refer them to foster care or adoption or them a way of conceiving themselves*”.(Participant 20, Social worker)

“*Also, if they want to have kids and then they are two males, adoption, because we do have people who say Sister, I don’t want this kid*”.(Participant 24, registered nurse)

#### 3.7.2. Other Fertility Technologies

Many HCPs highlighted discrete methods available for genderqueer individuals who desire to become parents. For example, most of the HCPs mentioned finding a sperm or egg donor, artificial insemination through in vitro fertilization (IVF), and using surrogacy as necessary services to be made easily accessible and affordable for genderqueer individuals.

“*I don’t know how to put it, like planning to have kids in vitro, IVF. Yes, IVFs. I know those kinds of needs. I know. Also, for some of them. Yeah. No, that’s what I can think of OK*”.(Participant 2, clinical psychologist)

“*Okay. With the gay people, Yoh, this is difficult for me. With the gay people I think their reproductive needs, again, I think it’s the same as the lesbian people. Like later stage when they wish to become parents, they can do artificial insemination as well*”.(Participant 10, Registered nurse)

“*You understand that it is possible nowadays that one can still have a baby, one can still find a sperm donor, and people are, some of them are for it. I’ve had a couple that had that, both girls they found a sperm donor, they had a child. These services are important for queer people but are expensive and scares (sic), I wish they could be available everywhere*”.(Participant 21, clinical psychologist)

### 3.8. Safe Availability of Intimacy Tools

Most of the HCPs cited sexual apparatus and instruments such as dildos and lubricants for genderqueer individuals as requisite SRHSNs that should be easily accessible. Moreover, the HCPs stated that availing these sexual instruments would enable genderqueer people to engage in safe sex. However, the HPCs emphasized the issue of health education related to cleanliness and hygiene during the usage of such sexual tools.

#### 3.8.1. Lubricants

The majority of HCPs indicated that genderqueer individuals should be provided with lubricants as part of their sexual needs. The HCPs related this to the anatomical structure of their particular sex organs, comparing it with a heterosexual woman’s sex organs, thus stating the importance of lubricants minimizing injuries. Further, the HCPs stated that lubricants should be easily accessible and affordable, like the contraceptives rendered in healthcare facilities.

“*Yah, you see, they want anal sex. So, well it’s not only about gays, even us straight people-So, well, every time when you want to try anal sex, remember it is not the same as vagina, it stretches but anus doesn’t. So, I think they need lubricants for their sexual engagement so that they don’t tear off*”.(Participant 15, registered nurse)

“*And then with our gay guys, also I would say we would promote safe sex with them by providing things such as lubricant. Yes, lubricants need to be given everywhere in the facility like we give them condoms*”.(Participant 18, registered nurse)

#### 3.8.2. Sexual Aids

The HCPs also mentioned that lesbians should be provided with sexual aids and be comprehensively informed about maintaining good hygiene with such devices to reduce the spread of sexual infections.

“*For a lesbian I think because… we just need to give information regarding… Remember, they use sex toys, right. We can just give information on how to use them, like the cleanliness of those toys, and then we just give any information. [Laughter] I don’t have much information about those toys, that’s the thing. But I think cleanliness will be number one for me*”.(Participant 10, registered nurse)

“*Lesbian when they come to the hospital, they need to be checked for STIs because of the things they use to have sexual intercourse […] Oh, the toys they use. So somewhere somehow sometimes they need to be given those things and need to be taught on how to take care of them nicely especially if ever they don’t know how to clean them*”.(Participant 15, Registered nurse)

“*If a lesbian comes in and talks about whatever, I think I will give her information from my own understanding, my mind will be telling me oh by the way they are saying lesbians are using fingers, they are using– what do they call them? Apparatus they are using and whatever, and meaning those things needs to be clean before they can be used. So, I will talk more about cleanliness*”.(Participant 31, Registered nurse)

### 3.9. Transition Navigation

Nearly all the HCPs revealed that some gender-fluid individuals might want to transition and change their gender. In addition, the HCPs acknowledged that transgender women and men specifically require different methods of gender transitioning services and indicated invasive procedures and medical treatments as essential healthcare needs and services for such individuals.

#### 3.9.1. Gender Reassignment Surgeries

Many HCPs said that medical procedures are services needed by genderqueer individuals who choose to change their birth gender and genitals. Moreover, the HCPs indicated that most genderqueer individuals required surgery for gender affirmation; however, they were still being judged, and their significant others found it difficult to accept and adjust since they had known them with their natal sex.

“*Also, the needs for some people I know the needs are for transitioning needs like gender reassignment surgeries or services*”.(Participant 2, clinical psychologist)

“*They need those services of changing their genitals, but people judge them by saying why are you changing, why are you are you doing this? So, it’s the experience that I had with them*”.(Participant 4, social worker)

“*Like people who want to change their gender. Let’s say maybe he is a male; he wants to be a female. I think they should consider giving them that opportunity to change their gender, yes, those procedures*”.(Participant 7, enrolled auxiliary assistant)

#### 3.9.2. Hormonal Interventions

Most HCPs indicated that non-invasive medical treatment was also a service needed by genderqueer individuals who desired to effect physical changes that aligned with their gender identity. The HCPs stated that hormone replacement therapy (HRT) for both transgender men and women is a healthcare need to assist them in developing secondary sex characteristics that are more congruent with their affirmed gender. However, some HCPs struggled to explain what transgender individuals needed, although they shared their experiences with transgender patients and hearsay reports that HRT is one of their crucial healthcare needs.

“*Members of the queer community, what I understand is that some of them they come for hormonal balancing and yes they need those hormones to change their gender and develop breast, Eish!!*”.(Participant 4, social worker)

“*Transgender ones, that’s a tricky one, I’ve never dealt with one. I’m not sure because what I know is that when you… the females apparently, they get your… is it testosterone or what, so the hormones are not functioning as a female anymore, so they get some shots apparently to change their hormonal system or your… what do you call it, that, I am not sure what to call it but I do hear that they do get your Depo-Provera for something that has to do with their hormones; I am not sure what but I’ve never dealt with one*”.(Participant 8, registered nurse)

“*I remember a client who came, and he is undergoing hormonal therapy to transgender from being a male to a female, and she was asking for hormones like your Depo injection, it’s part of the medication that he/she was taking for the hormonal change and all that*”.(Participant 32, medical doctor)

## 4. Discussion

Our research emphasizes the critical need for genderqueer-inclusive healthcare facilities. The participants of this research study stressed the significance of diversity and proposed specific solutions such as gender-neutral bathrooms and genderqueer-friendly services. The inadequacy of accommodations for genderqueer individuals in government healthcare facilities, notably in male and female bathrooms, was a significant source of concern among the HCPs. They voiced concern about the difficulties genderqueer individuals, particularly those who identify as “shemale”, face while deciding which facility to use. They also proposed offering clear indicators and guidance for genderqueer people to use disability bathrooms not labeled by gender. Our findings are consistent with those of Francis et al. [26], who find support for gender-neutral bathrooms; furthermore, in Francis et al. [26], policymakers and school staff expressed support for gender-neutral bathrooms, particularly as an approach to reducing the bullying of trans students, and other participants in the same study noted additional benefits of gender-neutral facilities, such as their potential to eliminate the self-consciousness trans students experienced. In addition, the literature reflects strong support and advocacy for gender-neutral bathrooms, emphasizing their usefulness in building a sense of acceptance and inclusivity [27,28]. However, other studies show that gay people face health issues such as depression and suicide [29]. Even though the study by Huff et al. [30] focused on gender-neutral bathroom availability across campuses in the United States (U.S.), our findings agree with their inferences that improving access to gender-neutral bathrooms will assist in reducing genderqueer individuals’ fear of harassment and psychological distress. Lastly, our research agrees with a study by Deutsch [31] that emphasizes the establishment of gender-neutral bathrooms for non-binary individuals as well as those who feel uncomfortable in binary, gendered surroundings. Other studies have demonstrated the use of some activities to ensure genderqueer inclusivity and affirmability; for example, a study by Lee [32] finds that queer identity signs and symbols are vital in assisting genderqueer individuals to feel safe in their workplaces. In addition, Hadland et al. [33] suggest strategies such as posting materials on walls indicating that a facility is queer-friendly, HCPs wearing rainbow pins, and the use of language that builds trust between genderqueer individuals and HCPs.

Our study’s findings demonstrate the importance of integrating psychological, counseling, and therapeutic care for genderqueer individuals into SRHSNs. Gender-fluid individuals’ diverse concerns demand a comprehensive strategy that covers not only physical components but also the emotional well-being of persons undergoing gender changes and those struggling with various sexual and reproductive health issues. Our data support the notion discussed in several studies, such as the findings by Heredia et al. [34] underscoring the necessity of developing integrated care models whereby mental health providers in primary care are particularly well positioned to provide affirming treatments to lesbian, gay, bisexual, and queer (LGBTQ) patients navigating the healthcare system. This report further stated that the effective integration of LGBTQ-affirmative behavioral health treatments in primary care necessitates knowledge of a wide range of biological, psychological, social, and cultural health issues affecting LGBTQ people [34]. Our study lends support to the idea that counseling psychologists could also strive to mainstream sexual and reproductive health (SRH) and related talks by paying particular attention to word selection and vocabulary that does not make assumptions [35]. Our findings are in agreement with the conclusions by Mitra and Globerman [36] that it is critical to build healthcare facilities that provide medical treatment, counseling, and referral services to lesbian, gay, bisexual, and transgender (LGBT) people to ensure access to services.

The participants in our study proposed holistic and customized methods for genderqueer sexual well-being. The HCP findings indicate a recognition of the various needs of genderqueer individuals and the significance of providing comprehensive treatment that covers both sexual and reproductive health problems. The proposed SRHSN, which includes breast examinations, male circumcision, pap smears, prostate services, and urological care, emphasizes a comprehensive and inclusive approach to genderqueer individuals’ sexual well-being. A few studies do not explicitly mention breast examinations, male circumcision, pap smears, prostate services, and urological care for genderqueer individuals’ SRHSNs, but they align well with our study. For example, a study by Herriges et al. [37] discusses the relationship between sexual orientation and prostate, breast, and cervical cancer screening and diagnoses. Furthermore, that study confirms the need for LGB people to be examined for cervical, prostate, and breast cancer, as their findings reveal that more heterosexual individuals are screened for these cancers than LGB people and that there is no association between sexual orientation and cancer diagnosis [37]. Another study finds that younger participants prioritized routine preventive care such as pap screenings, sexually transmitted illness prevention, and birth control [16]. Our findings support the notion that the long-term threat in transgender females receiving hormonal therapy is unknown; thus, the best possible treatment must include pretreatment education and assessment, individualized dosing, ongoing routine monitoring, and standard breast and prostate cancer screening [38]. This is confirmed by Patel et al. [39] who conclude that the tendency of steroid hormones to stimulate the growth of some tumors raises concerns regarding the safety of different doses and treatment combinations, based on the conclusion by Li et al. [40] that LGB cancer survivors have worse survivorship than their heterosexual peers, with heterogeneity in subgroups. Our study reaffirms the importance of the proposed holistic and customized methods for genderqueer sexual well-being, as evidenced by Mitra and Globerman [36], that HCPs might underestimate the risk of cervical cancer in lesbians, increasing the risk of some cancers. This finding is consistent with Bass and Nagy’s [2] study, which finds the widespread view among bisexual and lesbian women that pap smears and regular screening are unnecessary.

The participants advocated for increased accessibility and specific healthcare services suited to the reproductive needs of genderqueer individuals. Addressing the genderqueer individual’s unique issues and vulnerabilities, such as offering CTOP services and promoting safe sex practices through condom accessibility and education, leads to more welcoming and beneficial healthcare facilities. There is agreement between our findings and key recommendations reported in Rigilano [41] regarding offering contraceptive counseling and encouraging all family physicians to be attentive to this oft-neglected area of prevention in the population of women who have sex with women (WSW) because unintended pregnancy is a substantial problem among WSW. However, the challenge remains that lesbians do not use contraception. For example, a study by Charlton et al. [42] finds that lesbians are less likely to use any method of contraception and a study by Everett et al. [43] reports that lesbian women were less likely to receive contraceptive counseling at pregnancy tests, and lesbian women without male partners were less likely to receive condom use counseling during sexually transmitted disease-related healthcare visits than heterosexual women. According to our findings, lesbian and bisexual women are more likely to be raped than heterosexual women, extending the findings of Canan et al. [44] who also find a higher incidence of lesbian and bisexual women experiencing rape than heterosexual women. For example, Canan et al. [44] discovered that 63% of bisexual, 49% of lesbian, and 35% of heterosexual women had experienced rape in their lifetimes. Furthermore, a study by Bowler et al. [45] reveals that some genderqueer individuals seek out unsafe abortions. Our study supports the argument that heterosexuality should not be presumed and agrees with the findings by Hodson et al. [8], indicating that lesbian and bisexual women (LB) are more likely to have unplanned pregnancies and terminations.

Our study highlights the significance of enhancing services related to HIV prevention, PrEP access, and STI treatment for genderqueer individuals. The HCPs’ suggestions emphasize the significance of specialized and readily available healthcare services that address the specific risks and health needs of genderqueer individuals. According to our findings, providing comprehensive HIV prevention strategies, such as PrEP and PEP access, as well as STI information and treatment, will help create a more inclusive and supportive healthcare environment for genderqueer individuals. Our findings support the importance of the study by Dorcé-Medard et al. [46] which finds that PrEP is critical for limiting HIV transmission among the LGBTQ+ subgroup. Furthermore, our findings support and agree with the findings of Alcantar-Heredia et al. [47] that PrEP use is associated with a sense of belonging, trust, and security about their sexuality, as well as another study by Quinn et al. [48] that discovered four unexpected benefits of PrEP for young Black men who have sex with men (MSM): improved engagement in medical care, decreased sexual and HIV anxiety, raised sexual comfort and freedom, and positive sexual relationships with individuals living with HIV. Despite the importance and benefits of PrEP, HIV testing, and prevention among LGBT people, other studies have revealed challenges to SRH access in this location, such as transportation, a lack of medical insurance, and the expense of treatments or obtaining services without parental knowledge [49]. Again, our findings are consistent with previous research. Barriers to HIV and STI testing include fear of positive results, difficulty accessing sexual healthcare in general, service providers’ perceptions of low risk, a lack of HCPs’ knowledge, and limited clinic capacity to meet STI testing needs [50]. Our findings support the assertion made by O’Farrell et al. [51] that current sexual health education programs are primarily heterosexual, creating a sense of exclusion for LGBTQI+ youth. As a result, our findings support the idea that there should be chances to participate in peer-led sexual health programming and expanded HIV prevention campaigns dealing with women who have sex with women [52].

Our findings emphasize the necessity of acknowledging and assisting genderqueer individuals’ desire to become parents. Adoption and various fertility technologies are portrayed as empowering alternatives for genderqueer individuals, highlighting inclusivity and support on their path to becoming parents. The findings indicate a need for accessible and inexpensive options that address the genderqueer individual’s unique reproductive needs and goals. The results of our study provide supporting evidence for Bass and Nagy [2] that lesbian and bisexual women have special additional needs, i.e., that their needs are similar to those of all women; however, they face challenges such as the desire for childbearing and the need to discuss options for conception [2]. In addition, our study agrees with the conclusions drawn from a study by Violette and Nguyen [53] that lesbians are more likely to face difficulties with adoption, and only those with medical insurance are likely to seek assisted reproductive services and adoption services.

Although the findings of our study provide comprehensive and varied elements of the fertility technologies and challenges experienced by queer individuals regarding fertility, it is important to recognize fertility preservation (FP) as an additional method that queer individuals like lesbians and transmen could utilize. This is a technique of maintaining and safeguarding the reproductive tissues, sperm, and eggs for the purpose of having biological offsprings in future [54]. Studies have highlighted currently available, well-established FB technologies, such as embryo and oocyte cryopreservation, ovarian and testicular tissue cryopreservation, and sperm cryopreservation [54,55,56].

Other studies have also outlined how queer individuals used FP as their preferred method, where Brik et al. [57] state that 91% of queer adolescents were counseled on FP, only 38% of them tried FP, and 75% were able to cryopreserve sperm suitable for intracytoplasmic sperm injection. Chen et al. [58] showed that 4 out of 105 queer participants in their study completed sperm cryopreservation and one completed oocyte cryopreservation. Moreover, FP seems to have a low update; for example, a study by Riggs and Bartholomaeus [59] stated that very few respondents (7%) had engaged in FP.

By contrast, findings by Waalkes [60] indicate that both bisexual and lesbian women are less likely than heterosexual women to desire and intend parenthood and are more likely to consider adoption. However, bisexual women are more likely than both lesbian women and heterosexual women to express a desire for parenting unintentionally. Another study by Messina and D’Amore [61] reveals that while lesbians and gay men (LG) choose to adopt, they experience challenges with adoption, such as numerous self-doubts and emotional conflicts driven by introjected heteronormative assumptions about family. Furthermore, the study by Messina and D’Amore [61] indicates that during the adoption procedure, lesbians and gay men are confronted with a large number of difficulties and legal obstacles connected to their sexual minority status. Notwithstanding the challenges experienced by lesbians, gay men, and bisexual persons attempting to become parents, this study provides further confirmation of other studies demonstrating that conceiving using clinical fertility services is a healthcare need [62]; for example, the study by Power et al. [62] shows the factors influencing a parent respondent’s decision to use fertility services: 80% indicated access to donor sperm, 41% indicated fertility problems, and of those who had accessed donor sperm, more than half (57%) had used in vitro fertilization (IVF) services. Downing [63] shows that among all pregnancies using anonymous donor sperm, four in five were for women in same-sex relationships. Consistent with previous research by Iraklis [64], there is a need to avoid oversimplified and static explanations for lesbian women’s desire for parenthood and consider their choices within a broader, multifaceted context.

Our study underlines the importance of maintaining the safe availability of intimacy products like lubricants and sexual aids as part of a comprehensive SRHSN for genderqueer individuals. The HCPs campaign for accessibility, affordability, and education on the correct usage and hygiene of such devices and, further, the consideration of these instruments as essential components of the sexual well-being of genderqueer people. This facet highlights the significance of inclusive and knowledgeable healthcare practices that address the various needs of genderqueer individuals and, in addition, supports the established findings that MSM, bisexual men, and gay individuals use and require lubricants for their sexual activities. For example, a study by Lee et al. [65] finds that few men report that lubricants should be used to facilitate anal intercourse. Similarly, a study by Dodge et al. [66] finds that more than 90% of both gay and bisexual male participants report lifetime lubricant use, and lastly, a study by Tadele and Amde [67] finds that approximately two-thirds report using lubricants, while the percentages differ depending on the group: 68.3% (41/60) of gay people, 50.0% (8/16) of lesbians, and 70.5% (12/17) of bisexual people [67]. Our findings are consistent with the confirmed findings of a comprehensive review by Kennedy et al. [68] that most people approve of the use of lubricants for reasons of comfort/reduced discomfort and sexual pleasure. This is similar to the study by Dodge et al. [66], which finds that the most commonly cited reasons for lubricant use include increasing comfort during anal intercourse, curiosity, and having more comfortable sexual intercourse. Regardless of the benefits of using lubricants during sexual intercourse, our findings are similar to prior research [69], which finds that commodities for sexual and gender minority youth, such as dental dams and lubricants, are generally unavailable in public health facilities. Our findings correspond with the findings of other research finding that genderqueer individuals utilize sexual aids during intercourse [70,71,72], which indicates the importance of such devices as a specific SRHSN for individuals.

This study demonstrates the need to support genderqueer individuals’ differing gender reassignment requirements. The HCPs understood the need to deliver therapies such as gender reassignment surgeries and hormonal interventions to assist genderqueer individuals on their journey to gender affirmation. The difficulties genderqueer individuals experience through these transition experiences underline the need for a more inclusive and supportive healthcare environment for gender-fluid individuals. Our findings align with the literature, which indicates that hormone treatment therapy (HRT) and surgeries are the primary requirements for transgender men and women [16], as well as findings by Unger [73] indicating that many transgender individuals seek cross-sex hormone therapy for the treatment of gender dysphoria, and findings by Reisner [74] indicate that hormones and injectable silicone are two of the ten things transgender people should discuss with their healthcare providers. Another study by Akhavan [75] finds that gender-affirmation surgery reduces rates of gender dysphoria, depression, and suicidality while also significantly improving quality-of-life metrics. Our findings support the findings of Hadj-Moussa et al. [76] that gender affirmation surgeries (GASs) improve the quality of life, happiness, and sexual function of transgender people.

Lastly, our study underpins the need for and importance of HCPs’ continuous education and development regarding genderqueer healthcare matters. We believe this support could be achieved through continuous workplace workshops and in-service training to ensure HCPs remain up to date regarding genderqueer health and well-being matters. Similarly, Yu et al. [77] conclude that such ongoing training could enhance skills, knowledge, and changes in attitude toward genderqueer individuals. In addition, the literature shows that after training HCPs in genderqueer cultural competencies, there was a change in attitudes and increased knowledge [78,79].

## 5. Conclusions

This research study aimed to identify and explore genderqueer-specific SRHSNs among HCPs in Gauteng Province, South Africa. The data were collected from 33 participants in HCPs and qualitatively analyzed through thematic analysis, resulting in nine key themes.

Our research highlights the vital importance of establishing genderqueer-inclusive healthcare institutions. The lack of adequate accommodations, notably in gendered bathrooms, surfaced as a key problem among healthcare providers, prompting specific suggestions for gender-neutral facilities. The findings highlight the importance of including psychological, counseling, and therapeutic services in SRHSNs. The participants also emphasized the necessity of personalized sexual-reproductive education and comprehensive healthcare services, such as HIV prevention and addressing the unique reproductive requirements of genderqueer individuals.

This study calls for increased accessibility options, lower prices, and education on intimate products, emphasizing the importance of inclusive healthcare practices that serve the various needs of the queer community. Furthermore, our findings emphasize the significance of addressing gender reassignment needs, with healthcare practitioners understanding the necessity for inclusive interventions, such as gender reassignment operations and hormone therapy. Overall, these findings advocate for a more inclusive, supportive, and holistic healthcare environment for genderqueer people that recognizes and addresses their distinct physical and mental well-being. Our research recommends establishing genderqueer-inclusive healthcare standards to address the specific and unique sexual-reproductive healthcare requirements of genderqueer people. These standards should include the availability of gender-neutral facilities, sexual and reproductive health services customized to the LGBTQI+ community, and healthcare practitioner training programs to ensure responsiveness and competency in meeting the unique needs of genderqueer individuals. We acknowledge that bias is acquired in nature through qualitative studies; however, the principal investigator clarified the purpose of this research to all participants with an emphasis on improving knowledge in queer-related matters among HCPs before the commencement of data collection. Moreover, the principal investigator requested the participants to respond honestly based on their experiences, thus minimizing response bias from HCPs.

## Figures and Tables

**Figure 1 healthcare-12-01026-f001:**
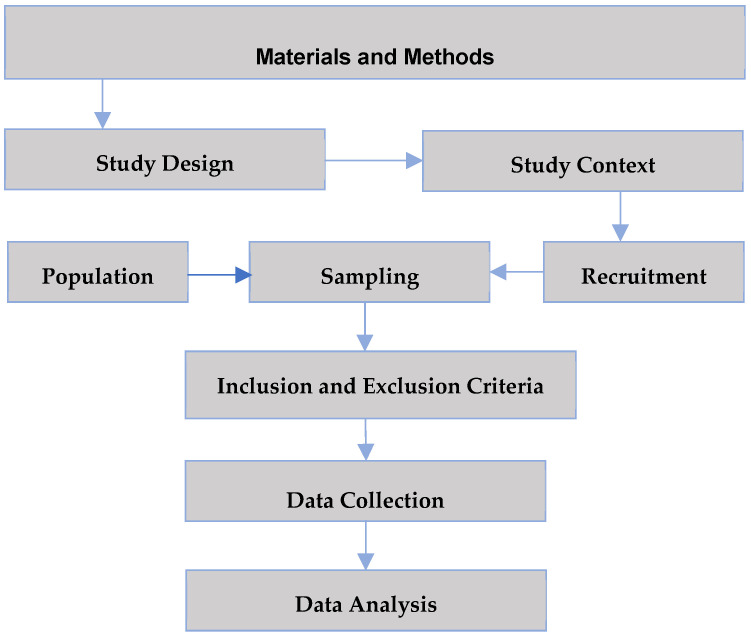
A flowchart of the workflow for the process of the methodology.

**Table 1 healthcare-12-01026-t001:** Sociodemographic data of the participants.

Participant No.	Age of the Participants	Sex of the Participants	Marital Status	Profession	Duration of Work as HCP
P1	40	Female	Single	Social Worker	16
P2	36	Female	Married	Clinical Psychologist	13
P3	23	Female	Married	Registered Nurse	3
P4	47	Female	Single	Social Worker	20
P5	47	Male	Married	Clinical Psychologist	25
P6	42	Female	Single	Registered Nurse	21
P7	32	Female	Single	Enrolled Nursing Assistant	6
P8	36	Female	Single	Registered Nurse	7
P9	57	Female	Single	Registered Nurse	36
P10	36	Female	Married	Registered Nurse	7
P11	25	Female	Single	Registered Nurse	5
P12	55	Female	Single	Registered Nurse	22
P13	42	Female	Single	Social Worker	12
P14	28	Female	Single	Medical Doctor	3
P15	43	Female	Married	Registered Nurse	17
P16	30	Male	Single	Registered Nurse	3
P17	24	Female	Single	Medical doctor	2
P18	32	Female	Single	Registered Nurse	4
P19	31	Female	Married	Enrolled Nursing Assistant	10
P20	29	Male	Single	Social Worker	5
P21	43	Female	Married	Clinical Psychologist	13
P22	42	Female	Single	Enrolled Nurse	16
P23	32	Male	Married	Medical Doctor	5
P24	44	Female	Single	Registered Nurse	13
P25	31	Female	Single	Registered Nurse	7
P26	37	Male	Single	Clinical Psychologist	15
P27	29	Female	Single	Registered Nurse	6
P28	27	Female	Single	Social Worker	4
P29	30	Female	Married	Clinical Psychologist	1
P30	37	Female	Single	Enrolled Nurse	10
P31	50	Female	Married	Registered Nurse	35
P32	28	Female	Single	Medical Doctor	6
P33	48	Female	Married	Clinical Psychologist	20

**Table 2 healthcare-12-01026-t002:** Themes and subthemes from HCP interview responses.

Themes	Sub-Themes
A crucial need for inclusive healthcare facilities	- Genderqueer-inclusive bathrooms.
The need for psychological, counseling, and therapeutic support in sexual and reproductive healthcare	
Access to sexual and reproductive education and integrating support	
Suggested reproductive health services for genderqueer people’s sexual wellness	- Breast examinations; - Male circumcision; - Pap smears; - Prostate services; - Vasectomy services;
Improved accessibility and particular genderqueer reproductive healthcare services	- Choice to terminate pregnancy; - Contraceptive accessibility and promotion.
Optimizing services related to HIV, PrEP access, and STI treatment	- HIV preventive measures and prophylaxis accessibility; - STI awareness and treatment.
Genderqueer people’s parenthood aspirations and empowerment	- Adoption options; - Other fertility technologies.
Safe availability of intimacy tools	- Lubricants; - Sexual aids.
Transition navigation	- Gender reassignment surgeries; - Hormonal interventions.

## Data Availability

The data are unavailable owing to the protection of the participants’ privacy and ethical constraints. However, it can be provided upon request and following authorization from SMUREC.

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
