# Peer review of "Holistic Sexual-Reproductive Healthcare Services and Needs for Queer Individuals: Healthcare Providers’ Perspectives"

_healthcare, 2024, doi:10.3390/healthcare12101026_

Round 1
Reviewer 1 Report
Comments and Suggestions for Authors
I have a number of difficulties with this paper that I think need attention or comment. (I have attached a pdf of the paper with questions raised and comments made throughout). Most notably, it is unclear whether the interview data provided in the paper in fact supports the claim that 'queer individuals' as an umbrella category have unique sexual and reproductive health needs and services. First, your participants often show hesitance around what they can think of that is unique when responding, I take it, to the two questions (which beg the question of the research): "what are the specific sexual-reproductive healthcare services and needs for queer individuals?”, and “what other specific sexual-reproductive healthcare services and needs do you consider important for queer individuals?” They often fall back to saying "like other men" or "as women" and in so doing seem to provide evidence contrary your hypothesis that queer individuals have unique SRHSN. Second, many of the 'themes' (needs) identified are not at all unique to individuals of the queer community--contraceptive use, cleanliness, termination of pregnancy, sexual and reproductive education, fertility treatments... none of these are unique SRHSN to the queer community--what may be unique here are the particular situations or contexts that give rise to these needs or within which these needs must be managed, but they are shared needs beyond the queer community. The first concern about restrooms shouldn't really count as a sexual and reproductive health need... should it? There are some unique concerns regarding hormonal treatments, and gender reassignment surgery, for instance, but these are not unique to the queer community as a whole, but rather to certain individuals within the queer community. Moreover, where these are discussed there is a significant conflation of gender and sexuality that needs to be addressed.
While I have difficulties with the research and the findings as discussed above, there are interesting findings in the data--notably preconceived and internalised patriarchal and heteronormative discourses reflected in the ways in which health practitioners speak about the SRHSN of the queer community. Indeed, many of these quotes point directly to the problematic discourses that foster the conditions in which different identities falling within the queer community are vulnerable to the kind of treatment the authors worry about. I suggest going back to the drawing board and looking at the assumption that there would be something unique to find in the first place, and the framing of the research in this way. I would then focus on the heteronormative discourse drawn on by HCPs in Gauteng and problematise this discourse and its place in reinforcing patriarchal heteronormativity (and the consequence of this in the lives of the queer community when it comes to accessing health care services.

This paper needs careful proofreading and extensive editing. It is littered with errors that are distracting to your reader.
Author Response
For research article
Response to Reviewer 1 Comments
- Summary
Thank you very much for taking the time to review our manuscript. We appreciate the positive feedback that will improve this manuscript. We have worked together as authors to ensure all the concerns, questions, advises, and recommendation made by you as reviewer 1 are addressed diligently. It was bit confusion since reviewer 1 did not classify the comments based on the different sections. We have looked at the attached PDF and we decided to group all the feedback based on the different sections in a table number 3. Please find the detailed responses below and the corresponding revisions/corrections highlighted in red in the in the re-submitted file.
- Responses from authors on general evaluation.
Questions for general evaluation |
Reviewer’s evaluation |
Response and revision |
Quality of English Language (This paper needs careful proofreading and extensive editing. It is littered with errors that are distracting to your reader. |
Extensive editing of English language required. |
Authors used professional editor (Shared editor’s certificate with the academic editor). We have attached an edited manuscript. |
Does the introduction provide sufficient background and include all relevant references? |
Must be improved. |
Improved as advised by reviewer 1, see the step-by-step comments below. |
Are all the cited references relevant to the research? |
Can be improved. |
See the step-by-step table for the comments. |
Is the research design appropriate? |
Can be improved. |
See the step-by-step table for the comments. |
Are the methods adequately described? |
Yes. |
No revision required. |
Are the results clearly presented? |
Must be improved. |
See the step-by-step table for the comments. |
Are the conclusions supported by the results? |
Must be improved. |
See the step-by-step table for the comments. |
- A table on point-by-point response to Comments and Suggestions for Authors.
SECTIONS |
COMMENTS |
RESPONSES BY THE AUTHORS |
Abstract. |
These are strange ways to describe queer individuals. Assumptions? |
Sentence was removed to avoid assumptions. |
Introduction. |
1. This is a truism. It does not need 'therefore' ahead of it as it is not the conclusion of an argument. 2. This of course applies to all, not just queer individuals. 3. Why are these behaviours unique to queer individuals? There are some old, conservative stereotypes expressed here. 4. It would be helpful for the authors to spell out these arguments, as they don't seem obviously wrong-headed from what I have read thus far. 5. Note, though, that this is a different concern to queer individuals having unique needs. If you are going to get this claim off the ground you could perhaps refer to certain treatments for trans individuals, but the authors are trying to be inclusive of the entire queer community, which doesn't seem to have SRHSN in common necessarily... At least this needs to be argued for, not assumed, or stipulated. 6. It needs to be shown that equal treatment is a bad thing... again, you need to argue for the specific needs you are suggesting queer individuals to have. 7. Such as? 8. WHY???? Why only queer individuals? 9. Not unique to queer people. 10. All these behaviours pertain to heterosexual and cisgender people as well. 11. Again, I find it offensive that these activities are deemed to only be practiced among the queer community. 12. This is definitely starting to look like an assumption that needs to be tested. I.e., it is a hypothesis and not fact as you have stated it here. 13. This sounds incredibly offensive. 14. Would it not have been wise to speak to queer individuals themselves about their SRHSN? |
1. “Therefore” was removed. 2. We agree, this was an emphasise to show some of the SRHSN which are important for queer individuals too. We decided to keep this to strengthen and show that referring to SRHSN these applies. 3. The title of our article is not about the uniqueness of queer individuals but holistic SRHSN for queer individuals which were identified by HCPs when asked about the specific SRHSN. We acknowledge these are not the unique, but the statement sentence is showing and emphasizing the importance of provision and inclusion of other services to improve the well-being of queer individuals. This sentence was not removed. 4. We appreciate the critique; nevertheless, we believe the manuscript appropriately conveys substantial arguments, supported by evidence, and while we are open to constructive change, we remain confident in the strength of our current presentation. No changes made. 5. We appreciate the critique; however, we also believe that literature or reference provided support arguments, the reviewer can clarify us on what they mean since there might be misunderstanding. No corrected made. 6. Corrected and argued with showing some specific needs of queer individuals “but looking at the other queer individuals do have need like gender transitions which include gender-affirming surgeries and hormonal therapy[16]. 7. Corrected and added one example “such as gender-affirming healthcare services, such as hormone therapy and gender-affirming surgeries”. 8. Because the study is on queer individuals, the rationale is to emphasize that if HCPs doesn’t know these SRHSN will be unable to render correct services to them, which will at the end affect queer. 9. Yes, but literature shows that they are affected. We do acknowledge that they are not unique, we cannot talk about heterosexual or cisgender in this section. (Authors 9 & 20 stated these on the queer. Are we supposed to disagree with what literature found. This is addressed similarly to number 3. No changes made as we believe that the statement emphasize on the gap. However, we have corrected the wording to avoid prejudice statements. 10. Same as number 9 comments, the findings are from author number 20. 11. Same as number 9 comments, the findings are from author number 20. 12. This comment is to show a gap and a need of the study. May the reviewer 1 advise on how to phrase this? We are showing that studies in SA regarding our aim were not done. To improve this, the reviewer could maybe suggest how to phrase it to avoid it being or sounding like a hypothesis. 13. This sentence was simplified and corrected to clear the aim/purpose of our article and was highlighted it red in the submitted document. “Hence, the purpose of this research was to identify and explore queer-specific SRHSN among HCPs in Gauteng Province, South Africa. The research aimed enhancing queer individual's access to SRHSN”. 14. Thank you for the suggestion. However, the population if our study/article was not on queer themselves. We will consider talking to them in the future studies. |
Material and Methods. |
1. Paper needs careful proofreading and editing. 2. You say 7 here and 8 below. 3. Swap the order of these two words. 4. Surely a question should have been, do you think there are any specific SRHSN that queer individuals have? 5. Editing required! |
1. As indicated, this paper was edited. 2. Corrected to “seven district”. This was an error. 3. Corrected to “seven district”. This was an error. 4. We acknowledge this comment, but our interview guide had open-ended questions only. What the reviewer 1 is suggesting is a closed-ended question. If HCPs said “No or Yes” then there were no going to be no data. Prior an interview, we had explained the purpose of the study to the participants, and they were aware of the study participants. 5. As indicated, this paper was edited. |
Results. |
1. Interesting statement from a psychologist revealing implicit biases ("lives like a queer"). 2. Why have you switched away from queer here? You have also shortened your label. 3. This is a crucial need for all people and is not, in itself, a need or service that is any more crucial for any one group. Education on particular concerns - such as transitioning, for example, yes - but education simpliciter is not more important for queer individuals. The paper needs to be rewritten with careful consideration given to the tacit meaning conveyed in the authors choice of words. 4. Editing required! 5. Not particular to queer individuals. 6. These are surely universal risks... or at least not risks only to the queer community. 7. This is key! 8. There is a tension between these two statements... sameness yet difference. 9. It seems to me that this quote speaks against the uniqueness claim for the umbrella 'queer' as the nurse here could also be read as indicating that she can't think of any unique services to offer lesbian women? 10. There is a lot to unpack here! Why would gay men not have genitals? Is there a conflation of sexuality with transgender here? 11. What does this mean? 12. I'm not sure you have understood this comment. 13. Deserve is a strange word to use... it has a number of connotations... why might they not deserve? There is an air of the heteronormative in the participants that should be highlighted if it is coming from them. As it stands it is ambiguous whether this is also the authors' view. 14. What does this mean? 15. Again, this can be read as emphasising sameness rather than specificity. 16. This is key. Does it not suggest that they are not sure that there are specific, unique needs that cover the umbrella, rather than particular identities. 17. Again, evidence that should make you doubt your hypothesis. 18. These are not needs that are unique though! 19. Qua woman not qua lesbian! 20. I would be interested in hearing more about this. 21. These are interesting contextual factors that make access to services challenging, and this would be something that would be interesting to explore, this said, the need for the termination is not unique. What is unique is the reason the termination is required in the first place. 22. Huge assumption. 23. Again, there is a conflation of gender and sexuality here. These internalised stereotypes would be worth exploring. 24. There is an assumption here that STIs (like HIV) are more common among queer people. This assumption needs to be interrogated. 25. Another huge assumption. 26. An interesting implicit bias to unpack here. 27. Again, an interesting moment that problematises your uniqueness hypothesis. 28. More evidence of the sameness rather than difference hypothesis. |
1. This seems like a comment/reviewer’s expression. No revision made. 2. Corrected “to queer individuals”. 3. We have replaced the word specific with holistic. This then aligns with the topic too. As explained above these were the findings from our study, regardless of the aim but HCPs/participants noted these are important for the queer individuals. 4. This paper was edited as advised and recommended by reviewer 1. 5. This got resolved after the removal of the word “specific” with “holistic”. 6. Same as number 5, it got resolved after the removal of the word “specific” with “holistic”. 7. This seemed to be an agreement of reviewer 1 on the participant quotation. No revision was done. 8. Corrected “HCPs emphasized that queer individuals deserve to be treated holistically to cater their SRHSN”. 9. The description was corrected” Many HCPs mentioned that lesbian women should be provided services and needs that are related to women well-being regardless of their sexual orientation such as contraceptives”. 10. Corrected the statement to avoid prejudice comments “However, some HCPs questioned whether gay men had genitals or not”. Additionally, reviewer 1 should note that these are the findings from the study, and we cannot change what participants believe or uncertainties. We have paraphrased it so that it be clear. 11. This a quotation from the participant (P16). Hence, in our description we showed that “some HCPs questioned whether gay men had genitals or not”. And still reviewer 1 questions it. That’s the participant’s quotation and we cannot change what was not said by us, this will insinuate false/fake data. No revision was made but we have provided clarity. 12. Clarity was provided on number 11. 13. Edited and corrected” HCPs noted pap-smear as one of the vital SRHSN for queer individuals and focused mainly on transgender and lesbian women. HCPs acknowledged that transgender and lesbians’ individuals should be examined and excluded from diseases such as cancer and indicated that conduction of the pap-smear examination will help them in preparation for fertility”. 14. Corrected, the words “akere o” were removed and proper English words used “yes pap-smear, if the client is”. 15. Corrected and paraphrased “A gay patient, for sexual and reproductive… because he is a man, right? So, they’ve got prostate, at times they have got urine tract infections, they should be examined” (Participant 9, Registered nurse)”. 16. Yes, they did not know about the specific and unique SRHSN of queer individuals, hence, we justified that they provided holistic SRSHN regardless of them being asked if they know the specific SRHSN. This sub-theme falls under holistic approaches; hence, they suggested urology and vasectomy services. Again, we aimed at identifying and exploring the specific SRHSN of queer, but findings came as general that is why we used the word holistic while some HCPs noted specific. 17. This issue has been clarified many times regarding the study, aim, and findings. We cannot change the participants quotation but meaning out of them. No changed made. 18. This was corrected and edited by providing the word “comprehensive” which removed “specific”. This provides clarity on the needs being comprehensive based on the participants and findings. 19. Corrected with the word” queer women” 20. Nothing to provide, it was paraphrased to avoid authors prejudice statements. 21. We have noted the feedback, since our study was not on factors to access services, we made no revisions. Regarding the matter of uniqueness has been addressed and corrected throughout the document where applicable. 22. Corrected and the paragraph was edited “HCPs illustrated that contraceptives are also lesbian’s needs, oral contraceptive pills and implants were stipulated as some of the contraceptive that can help them stop and control their menstrual cycle, thus making them feel comfortable as lesbians in their sexual orientation”. This was simplified thus avoid assumption stated by reviewer 1. 23. This seemed to be reviewer 1 ‘s reflection on the participant’s quotation (P22). In the near future we shall look in to factors and internalised stereotypes as advised but not for the purpose of this paper as the aim would not be relevant. 24. This feedback is the same as number 23. The advice is noted and shall in the future be looked into as advised but not for the purpose of this paper as the aim would not be relevant. 25. Since the reviewer was questioning the description of the authors based on the participant’s feedback. We added the exact word that was said by participant 32 to show that we were not making up statements and assumptions, but these are the findings shared by P32. We added “Adventures” removed the synonym exploring in nature, thus avoiding our descriptions to be prejudice. 26. This seemed to be reviewer 1 ‘s reflection on the participant’s quotation (P13). 27. Addressed lot of times about the issue of uniqueness and holistic. However, in this case it shows how P13 views, and we believe that this is one of the comprehensive needs for the queer individuals. In the overall or broader theme, we never mentioned uniqueness. Yes, the study wanted to find out those. This was about balancing representation of the data, including negative or contradictory information. 28. Addressed lot of times about the issue of uniqueness and holistic. However, in this case it shows how P15 views, and we believe that this is one of the comprehensive needs for the queer individuals. In the overall or broader theme, we never mentioned uniqueness. Yes, the study wanted to find out those. This was about balancing representation of the data, including negative or contradictory information. |
Discussion |
1. This should be expanded on. 2. None of these are unique 3. I think the assumptions and biases uncovered in talking about this topic are worth exploring. 4. This needs more nuancing! |
1. This is not clear how we should expand it. The statement is from literature and forms part of the emphasis or emphasising the above paragraph. At least, the reviewer should indicate what they mean on the issue of expansion. We can then correct it. 2. We acknowledge that none of those SRHSN mentioned are not unique. As indicated in the introduction section number 3, that the title of our article/study is not about the uniqueness of queer individuals but holistic SRHSN for queer individuals which were identified by HCPs when asked about the specific SRHSN. We acknowledge these are not the unique, but the statement sentence is showing and emphasizing the importance of provision and inclusion of other services to improve the well-being of queer individuals. Yes, the aim was to identify the specific SRHSN, but HCPs ended providing what they believe is unique and the holistic ones. We went through our document and replaced the work unique and specific with holistic where applicable. This sentence was not removed. 3. We went through our document and replaced the work unique and specific with holistic where applicable. 4. The term “gay” was replaced with queer for consistency and to cater all individuals. This is the statement that was added “some of the queer individuals results in practicing”. |
Conclusion |
No comments from the reviewer. |
No revision required. |
Declaration |
No comments from the reviewer. |
No revision required. |
List of references |
No comments from the reviewer. |
No revision required. |
Reviewer 2 Report
Comments and Suggestions for Authors
The article brings a relevant topic presented in a well-structured way. It describes the needs and specific health care in sexual and reproductive health of the queer population, talks about health professionals and their performance by compulsory hetero and cisnormativity, and highlights the need of training unprepared professionals in aspects of sexual and reproductive health of the mentioned population groups.
Suggestions for the article:
In general, I suggest reevaluating the use of terms that may reinforce prejudices about the behavior and health of the queer population, for example, mentioning risky sexual behavior in a definitive way such as "poor sexual practices", bringing a load of judgment into the words.
In the introduction, I suggest revising the description of the objective at the end of the introduction, into a simple sentence. I also suggest bringing the justification in another sentence.
In the methodology, I suggest describing the type of study as described in the abstract. Also adjusting what is the main focus of the research as mentioned at the end of the introduction that mentions that the objective is to understand the perception of health professionals. At the beginning of the method in the Study Design section it already mentions that the purpose is to understand the needs of health of the queer population to develop an application “mHealth descriptive approach”, which is not mentioned in any other topic in the article.
In the results, I suggest caution with the use of words that could reinforce prejudices about the behavior and health of the queer population in paragraphs that do not describe the textual speech of health professionals.
I suggest reviewing Table 1, some professionals have a Duration of work as HCP incompatible with their age, such as P3.
In the discussion, I suggest emphasizing the need for continuing education and the development of a critical view of health professionals.
I emphasize that a non-inclusive environment is not just the issue of the bathroom for all genders and sexual orientations, it is important to point out that there may be symbols, flags, or other items that refer to a welcoming environment, even the body language of professionals, the vocabulary that is not appropriate and the way they target people, from reception to service.
In Line 808, I suggest reevaluating the sentence: “The aim of this research study was to was to identify and explore queer-specific SRHSN among 808 healthcare providers (HCPs) in Gauteng Province, South Africa”. In place of the word towas, wouldn't "was to identify" be more appropriate?
In conclusion, I suggest reviewing sentence repetition.
Comments on the Quality of English LanguageI suggest reevaluating the use of terms that may reinforce prejudices about the behavior and health of the queer population.
Author Response
For research article
Response to Reviewer 2 Comments
- Summary
Thank you very much for taking the time to review our manuscript. We appreciate the positive feedback that will improve this manuscript. We have worked together as authors to ensure all the concerns, questions, advises, and recommendation made by you as reviewer 2 are addressed diligently. Please find the detailed responses below and the corresponding revisions/corrections highlighted/in track changes in the re-submitted files.
- Responses from authors on general evaluation.
Questions for general evaluation |
Reviewer’s evaluation |
Response and revision |
Quality of English Language (I suggest reevaluating the use of terms that may reinforce prejudices about the behavior and health of the queer population.) |
Minor editing of English language required. |
Authors used professional editor (Shared editor’s certificate with the academic editor). |
Does the introduction provide sufficient background and include all relevant references? |
Can be improved. |
Improved as advised by reviewer 2, see the step-by-step comments below. |
Are all the cited references relevant to the research? |
Yes. |
No revision required. |
Is the research design appropriate? |
Yes. |
No revision required. |
Are the methods adequately described? |
Can be improved. |
Improved as advised by reviewer 2, see the step-by-step comments below. |
Are the results clearly presented? |
Can be improved. |
Improved as advised by reviewer 2, see the step-by-step comments below. |
Are the conclusions supported by the results? |
Yes. |
No revision required. |
- Point-by-point response to Comments and Suggestions for Authors.
Comment 1: In general, I suggest reevaluating the use of terms that may reinforce prejudices about the behavior and health of the queer population, for example, mentioning risky sexual behavior in a definitive way such as "poor sexual practices", bringing a load of judgment into the words.
Response 1: Thank you for a good observation. Authors worked together as a team and with the editor to check all terms that may reinforce prejudices. We went throughout the document and replaced the following: The word “poor” changed to “risky” on number 73.
Comment 2: In the introduction, I suggest revising the description of the objective at the end of the introduction, into a simple sentence. I also suggest bringing the justification in another sentence.
Response 2: We acknowledge the comment. We have revised the description of the objective at the end of the introduction. It was written in simple sentence “identify and explore queer-specific SRHSN among HCPs in Gauteng Province, South Africa.”. We then brought the justification on its own sentence just after the objective “The research aimed enhancing queer individual's access to SRHSN and enabling HCPs to better serve queer individuals with their specific and unique SRHSN, thereby improving and promoting their overall well-being”.
Comment 3: In the methodology, I suggest describing the type of study as described in the abstract. Also adjusting what is the main focus of the research as mentioned at the end of the introduction that mentions that the objective is to understand the perception of health professionals. At the beginning of the method in the Study Design section it already mentions that the purpose is to understand the needs of health of the queer population to develop an application “mHealth descriptive approach”, which is not mentioned in any other topic in the article.
Response 3: Thank you for noting this so that our study can be consistent. We have removed ” This was an exploratory qualitative research design to explain the queer’s specific sexual-reproductive healthcare services and needs (SRHSN) with the main purpose of developing a mHealth application that will be utilized by Healthcare Providers (HCPs) in Gauteng province to address the SRHSN of queer individuals” from the methodology and we have added “This was an exploratory sequential mixed method study, and the focus of this article is on the qualitative findings of the investigation.”. Indeed, there was a confusion regarding with “develop an application & mHealth descriptive approach” both were removed, and the sentence was corrected and the same objective from the end of the conclusion was added.
Comment 4: In the results, I suggest caution with the use of words that could reinforce prejudices about the behavior and health of the queer population in paragraphs that do not describe the textual speech of health professionals.
Response 4: Thank you for the feedback, it is appreciated, and this comment was an eye opener for the authors to avoid statements that are prejudiced, as might seem like they are supporting statements or quotations by HCPs. We have revisited and reviewed our result section to correct all words that could reinforce prejudices and we have highlighted the changes in red on the manuscript. Here are the changes made” Shemale changed to lesbians, words which were not in English such as “akere, di, ama” were removed, changed” need to take care of themselves and adopt protective measures” with” should practice safe sexual activities and adopt protective measures”. Again, we have removed “were not sure if gay men had private parts or not” with “However, some HCPs questioned whether gay men had genitals or not”. We have removed “have a vagina they”. We have replaced “since they are men certain assessments had to be conducted on them. For example, HCPs indicated that since gay patients are men and have prostate, they are worth of being checked to” with “they should also receive prostate services to”. Further, we have removed “were not sure of the types and specific SRHSN should be rendered to the queer individuals. However, they further”. We removed “’s vulnerability and inquisitively of trying new things some ended up” and added “can be at risk”. We have removed “due to their personal behavior and physical appearance”. We have removed “having and added “practice & and protecting themselves”. We have removed “since they do not wish to be fall pregnant intentionally or as a mistake”. We have removed “because they cannot bear and added “if they do not wish to bear”. We have removed “than using foreign objects that can hurt them and cause infections”. We have removed “help gay men from anal and added minimize”. We have removed “This came out as HCPs understand and acknowledges sex toys and their cleanliness as specific lesbian’s SRHSN”. We have added examples to provide clarification to the readers “such as contraceptives, and some.”
Comment 5: I suggest reviewing Table 1, some professionals have a Duration of work as HCP incompatible with their age, such as P3.
Response 5: Thank you, this was an error. We have checked with our raw data and corrected P3 duration of work “3”. We have rechecked all participants and noticed all were correct.
Comment 6: In the discussion, I suggest emphasizing the need for continuing education and the development of a critical view of health professionals.
Response 6: We have added a paragraph at the end of our discussion section that emphasizes the importance of HCPs to be trained and supported with workshops and Inservice training towards queer individuals matters. We added three studies to strengthen this need. (We have highlighted new references in the list in red 77-79). “Lastly, our study underpins a need and the importance of HCPs’ continuity education and development regarding queer healthcare matters. We believed that this support could be done through continuous workplace workshops and Inservice trainings to keep HCPs up-to date regarding queer health and well-being matters. Similarly, Yu et al (2023) concluded that could enhance skills, knowledge, and change of attitude towards queer individuals. Additionally, literature showed that after training HCPs in queer cultural competencies there was a change of attitudes and an increase on knowledge (Donisi et al (2020) &Lee et al, 2021)”.
Comment 7: I emphasize that a non-inclusive environment is not just the issue of the bathroom for all genders and sexual orientations, it is important to point out that there may be symbols, flags, or other items that refer to a welcoming environment, even the body language of professionals, the vocabulary that is not appropriate and the way they target people, from reception to service.
Response 7: We agree and acknowledge this feedback. It has helped use to add a brief statement from literature” Other studies have shown that the use of were some of the activities to ensure queer inclusivity and affirmability, for example a study by Lee (2023) showed that queer identities signs and symbols were vital in assisting queer individuals feel safe in their workplace. Additionally, suggested strategies such as posting materials on the wall indicating that it is a queer-friendly, HCPs wearing rainbow pins, and the use of the language that builds trust between queer individuals and HCPs”.
Comment 8: In Line 808, I suggest reevaluating the sentence: “The aim of this research study was to was to identify and explore queer-specific SRHSN among 808 healthcare providers (HCPs) in Gauteng Province, South Africa”. In place of the word towas, wouldn't "was to identify" be more appropriate?
Response 8: Thank you. This was a typing error and we have corrected it to “was to”
Comment 9: In conclusion, I suggest reviewing sentence repetition.
Response 9: Sentences were checked together with the editor to review if there are any repetitions. None noted. However, the manuscript has been edited.
Reviewer 3 Report
Comments and Suggestions for Authors
Dear Authors,
I read with great interest your Manuscript entitled “Holistic sexual-reproductive healthcare services and needs for 2 queer individuals: healthcare providers' perspectives.” which falls within the aim of this Journal.
In my honest opinion, the topic is interesting enough to attract the readers’ attention. Methodology is accurate, and conclusions are supported by the data analysis. Nevertheless, authors should clarify some points and improve the discussion citing relevant and novel key articles related to the topic.
Authors should consider the following recommendations:
- The manuscript should be streamlined in some paragraphs, such as in the discussion and conclusions, in order to improve readability.
- Fertility preservation options for queer individuals should be thoroughly discussed, especially in the female ones aroung the age of thirty-five. Please improve the discussion adding relevant references (refer to PMID: 37374095).
Comments on the Quality of English Languagenone
Author Response
For research article
Response to Reviewer 3 Comments
- Summary
Thank you very much for taking the time to review our manuscript. We appreciate the positive feedback that will improve this manuscript. We have worked together as authors to ensure all the concerns, questions, advises, and recommendation made by you as reviewer 3 are addressed diligently. Please find the detailed responses below and the corresponding revisions/corrections highlighted/in track changes in the re-submitted files.
- Responses from authors on general evaluation.
Questions for general evaluation |
Reviewer’s evaluation |
Response and revision |
Quality of English Language |
Minor editing of English language required. |
Authors used professional editor (Shared editor’s certificate with the academic editor). |
Does the introduction provide sufficient background and include all relevant references? |
Yes. |
No revision required. |
Are all the cited references relevant to the research? |
Must be improved. |
We have improved our citing by adding other references relating to fertility preservation. |
Is the research design appropriate? |
Yes. |
No revision required. |
Are the methods adequately described? |
Yes. |
No revision required. |
Are the results clearly presented? |
Yes. |
No revision required. |
Are the conclusions supported by the results? |
Yes. |
No revision required. |
- Point-by-point response to Comments and Suggestions for Authors.
Comment 1: The manuscript should be streamlined in some paragraphs, such as in the discussion and conclusions, in order to improve readability.
Response 1: Thank you for pointing out this issue. We have worked closely with the editor to ensure that paragraphs of the discussion and conclusions are simplified and easily readable. We have attached the edited manuscript.
Comment 2: Fertility preservation options for queer individuals should be thoroughly discussed, especially in the female ones aroung the age of thirty-five. Please improve the discussion adding relevant references (refer to PMID: 37374095).
Response 2: This was one of the most significant comments, we have looked at the referred article and added three more to discuss thoroughly fertility preservation options for queer individuals. Our discussion was mainly around supporting studies that emphasizes fertility preservation as one of the methods lesbians and transman can use.
Reviewer 4 Report
Comments and Suggestions for Authors
- The article focuses on the sexual-reproductive health needs of queer individuals from the perspectives of healthcare professionals, addressing an important and underrepresented issue in health research. For this purpose, it uses a mixed-method approach that provides greater understanding through qualitative data, making it suitable for examining complex and nuanced topics like LGBT health.
Here are my observations:
- The fact that the study included only healthcare professionals from Gauteng Province may limit the applicability of the findings to other areas or countries with different healthcare systems and cultural perspectives on LGBT people.
- A more detailed discussion of the methodological decisions made in the manuscript, specifically regarding how participants were chosen within healthcare settings, would be beneficial. It mentions purposive sampling but does not detail why specific providers were chosen, which could lead to selection bias.
Some necessary elements of qualitative research are still missing in this study:
- Description of the personal characteristics of researchers and interviewers, such as education and experience, that may influence their interaction with participants.
- Reflections on the possible relationships, power dynamics, or experiences that researchers may have with participants, affecting data collection and interpretation.
- Detailed description of the context in which the research was conducted, including geographical, cultural, or infrastructure aspects that may influence access to health services by queer individuals.
- More detailed descriptions of participants, including demographic data that may influence their perspectives on the sexual-reproductive health needs of queer individuals.
- Justifications for the number of participants and how they were specifically selected and recruited for the study.
- More detailed descriptions of how data was collected, including where interviews were conducted, how they were conducted, and any adaptations made to ensure the safety and comfort of participants.
- Detailed information on data analysis, such as code formation, theme development, and how conclusions were drawn from the data.
- Discussion on the verification and validation of the collected data, such as triangulation, checking with participants, or discussion among researchers.
- Balanced representation of the data, including negative or contradictory information that may have arisen during the research raised by participants or researchers.
- Although thematic content analysis is acceptable for qualitative data, the article should clarify how themes were extracted from the data. The analysis would be more credible if it included examples of code development and how these codes led to the final themes.
In the results:
- It is necessary to include participant quotations to support the found themes, ensuring an authentic representation of the participants' voices.
- If applicable, describe the software used for data analysis, explaining how it was used to organize or analyze the data.
- There is an underlying assumption that LGBT people have similar desires and challenges. Recognizing the diversity within the gay community itself—including the variety of experiences depending on socioeconomic class, gender identity, and race—could enhance the study.
- The data may be impacted by the biases of some healthcare providers, especially if they have preconceived notions about people who identify as LGBT. How these biases were addressed or mitigated during the interviews and analysis could be addressed in the publication.
Author Response
For research article
Response to Reviewer 4 Comments
- Summary
Thank you very much for taking the time to review our manuscript. We appreciate the positive feedback in improving this manuscript. We have worked together as authors to ensure all the concerns, questions, advice, and recommendation made by you as reviewer 4 are addressed diligently. Please find the detailed responses below and the corresponding revisions & corrections highlighted in red re-submitted files.
- Responses from authors on general evaluation.
Questions for general evaluation |
Reviewer’s evaluation |
Response and revision |
Quality of English Language (English language fine. No issues detected). |
English language fine. No issues detected. |
No revision required. |
Does the introduction provide sufficient background and include all relevant references? |
Must be improved. |
Improved as advised by reviewer 4, see the step-by-step comments below. |
Are all the cited references relevant to the research? |
Must be improved. |
Improved as advised by reviewer 4, see the step-by-step comments below. |
Is the research design appropriate? |
Must be improved. |
Improved as advised by reviewer 4, see the step-by-step comments below. |
Are the methods adequately described? |
Must be improved. |
Improved as advised by reviewer 4, see the step-by-step comments below. |
Are the results clearly presented? |
Must be improved. |
Improved as advised by reviewer 4, see the step-by-step comments below. |
Are the conclusions supported by the results? |
Must be improved. |
Improved as advised by reviewer 4, see the step-by-step comments below. |
- Point-by-point response to Comments and Suggestions for Authors.
Comment 1: The fact that the study included only healthcare professionals from Gauteng Province may limit the applicability of the findings to other areas or countries with different healthcare systems and cultural perspectives on LGBT people.
Response 1: Thank you for a good observation, we appreciate it. We think that if other scholars would intend to replicate the similar study in other parts of South Africa, Africa, or globally would focus on the methodology section more as the steps are applicable and can be replicated. We acknowledge that due to the differences of healthcare system, cultural perspectives, and other attributes across different regions the findings might differ. Thank you.
Comment 2: A more detailed discussion of the methodological decisions made in the manuscript, specifically regarding how participants were chosen within healthcare settings, would be beneficial. It mentions purposive sampling but does not detail why specific providers were chosen, which could lead to selection bias.
Response 2: We acknowledge the comment and we have added more details as suggested by reviewer 4. We have revised the description of our sampling method in the manuscript “We chose this method because HCPs have experiences in caring for different types of patients including queer individuals. Therefore, they were the relevant population to be purposively sampled due to their expertise and we anticipated them providing rich and detailed insights regarding the specific SRHSN of queer individuals. We have used a systematic approach to choose and select HCPs who rendered SRH to different patients in the healthcare facilities as we anticipated that they might be having experiences and encountered provision of SRH to queer individuals too”. See more details below the added paragraph on how recruitment occurred.
Some necessary elements of qualitative research are still missing in this study:
Comment 3: Description of the personal characteristics of researchers and interviewers, such as education and experience, that may influence their interaction with participants.
Response 3: Thank you for noting this so that our study. We have added a brief description of the principal researcher characteristic “The study was conducted with one principal investigator and the research assistant, who holds master’s degree in public health. The researcher received refresher training on how to conduct in-depth interviews using an interview guide for data collection from main supervisor during the master’s studies and prior the data collection of the data of this study. The principal investigator acquires knowledge on qualitative studies through students’ supervision, this enhanced in-depth interactions with participants. The research assistant holds honors degrees in industrial psychology and trained by the principal investigator on data collection steps and on conduction of interviews”.
Comment 4: Reflections on the possible relationships, power dynamics, or experiences that researchers may have with participants, affecting data collection and interpretation.
Response 4: We appreciate the reviewer's insightful feedback regarding the potential impact of researchers' relationships, power dynamics, and experiences with participants on data collection and interpretation. While we acknowledge the importance of these factors in qualitative research, it is important to note that our study did not explicitly explore these aspects. Moving forward, we recognize the value of reflecting on and addressing researchers' roles and interactions within the research process. While these reflections were not included in the current study, we believe they are vital considerations for future research endeavors in this area.
Comment 5: Detailed description of the context in which the research was conducted, including geographical, cultural, or infrastructure aspects that may influence access to health services by queer individuals.
Response 5: Thank you, the noting, this has helped us to add a brief sentence to emphasize the context of our study “These district hospitals get outreach as well as backing from general specialists stationed in regional healthcare facilities, implying that they are typically smaller institutions that get support and help from general specialists working in bigger regional hospitals”. However, this is not a limitation for HCPs to provide SRHSN to all patients including queer individuals as these are basic services rendered, if services are not offered in the district hospitals HCPs should transfer the clients.
Comment 6: More detailed descriptions of participants, including demographic data that may influence their perspectives on the sexual-reproductive health needs of queer individuals.
Response 6: Thank you so much for the feedback. We have added a brief under the result section just after a table for participants to provide a more detailed descriptions of participants as the table mainly summarized their demographics “Our study included 33 HCPs queer participants, with five identifying as male and 28 as female. The participants’ age mean was 36.7. When questioned about their marital status, 22 were single and 11 were married. All of them had tertiary education and were HCPs; five social workers, six clinical psychologists, 17 nursing staff with different categories, and four medical doctors. The participant with longer duration of work was one with 36 years, the middle duration of work was 25 years, and lower duration of work was one year”.
Comment 7: Justifications for the number of participants and how they were specifically selected and recruited for the study.
Response 7: Thank you for the feedback. This information was already included, reviewer 4 might have missed it. Please check 124-133 we discussed the recruitment process and 121-124 for justifying the selected participants. However, we have added a sentence which will help the reader in terms of justification and guidance of our study setting “The study population was guided by data saturation, which means that the principal investigator stopped collecting data once no new information was coming up from the participants”.
Comment 8: More detailed descriptions of how data was collected, including where interviews were conducted, how they were conducted, and any adaptations made to ensure the safety and comfort of participants.
Response 8: Thank you for the feedback. We have provided areas where the reviewer 4’s questions were already addressed in the initial submission. We understand this might have been missed by mistake. (1) How was data collected: number 152-180, The process was provided in 152-180, (2) Where interviews were conducted, was provided in, number 172-174, (3) How interviews were conducted was provided in, 172-180 (4) Any adaptions, we had different adaptions that we ensured, please see number; 159-166 adaptions of initial interview guide, 239-246, for privacy and confidentiality.
Comment 9: Detailed information on data analysis, such as code formation, theme development, and how conclusions were drawn from the data.
Response 9: Thank you for the feedback, this has helped us to add some of the activities done when analyzing our data thus will enable the replication the reviewer 4 asked earlier. We have reworked on our section to provide more clarity “The first coding step was used to classify and categorize the different codes and the principal investigator shared the first coding with one supervisor who acted as a peer reviewer to confirm the creation of codes. Additionally, the principal investigator developed a manual codebook which was later imported into NVivo V12. NVivo 12 is a software used for qualitative data, it enables the researchers to import transcripts, create themes, coding, and collaboration between students and supervisors. The second coding step commenced with clarification and grouping similar codes to develop themes, this was first done by the principal investigator and confirmed by supervisors. Finally, themes were developed, whereby the researcher grouped codes from second coding with similar meanings and interpreted them into full meaningful phrases thus making them more concise. Conclusions were drawn from the direct quotations from participants without changing their meanings.
Comment 10: Discussion on the verification and validation of the collected data, such as triangulation, checking with participants, or discussion among researchers.
Response 10: Thank you so much, this question was answered in comment 9 as we have reworked on our sentences where supervisors acted as peer reviewers to validate and confirm the collected data, was also started during pilot study until the end. We appreciate your feedback.
Comment 11: Balanced representation of the data, including negative or contradictory information that may have arisen during the research raised by participants or researchers.
Response 11: Thank you for the feedback, which makes us happy to hear that there should be balanced information in the presentation. We are glad as other reviewers saw it as something which is not acceptable. We are very glad. However, based on our findings there wasn’t much to contraindicate with but in our results presented we have shown how other HCPs felt and responded, please see what was included in number”355-356, 393”.
Comment 12: Although thematic content analysis is acceptable for qualitative data, the article should clarify how themes were extracted from the data. The analysis would be more credible if it included examples of code development and how these codes led to the final themes.
Response 12: Thank you this was amended and answered in comment 9. If reviewer 4 would like to see examples, we can add them but for now as authors we have agreed that the description in comment 9 will guide in terms of replication and credibility. We avoided having an over congested manuscript. Please feel free to contact us regarding this matter.
In the results:
Comment 13: It is necessary to include participant quotations to support the found themes, ensuring an authentic representation of the participants' voices.
Response 13: Thank you, we have added additional quotations to ensure the support of the themes as suggested by reviewer 4, thus, ensuring an authentic representation of the participants’ voices. Please see them in red under relevant themes in the resubmitted manuscript.
Comment 14: If applicable, describe the software used for data analysis, explaining how it was used to organize or analyze the data.
Response 14: Thank you, this was addressed with comment 9. We have briefly provided our understanding of NVivo 12, and its uses.
Comment 15: There is an underlying assumption that LGBT people have similar desires and challenges. Recognizing the diversity within the gay community itself—including the variety of experiences depending on socioeconomic class, gender identity, and race—could enhance the study.
Response 15: We appreciate the reviewer's valuable feedback concerning the diversity within the LGBT community and its implications for our study. While we regret that our completed study cannot be altered to address this concern, we acknowledge the limitation it presents. Moving forward, in our future studies we will explore the intersectional nature of LGBT experiences. We hope that our study, despite its limitations, can contribute to ongoing discussions and inspire further exploration of diversity within the LGBT community.
Comment 16: The data may be impacted by the biases of some healthcare providers, especially if they have preconceived notions about people who identify as LGBT. How these biases were addressed or mitigated during the interviews and analysis could be addressed in the publication.
Response 16: Thank you for noting this and we have included a brief paragraph on how bias was avoided in our study under conclusion section” We acknowledge that bias is acquired in nature through qualitative studies, however, the principal investigator clarified the purpose of the research to all participants with an emphasis of improving knowledge of queer related matters among HCPs before commencing so that can data collection. Moreover, the principal investigator requested the participants to respond in honestly based on their experiences thus minimizing response biasness from HCPs.”
Round 2
Reviewer 1 Report
Comments and Suggestions for Authors
This version of the manuscript is much improved. The careful editing of the manuscript allowed me to engage with the study in a way I found difficult to before. I think the focus on holistic SRHSN addresses many of my concerns in the first review over what I thought was an emphasis on the unique nature of these needs and services. Of course, some of them are unique, but many are not, and the move to holistic helps me deal with that concern.
I thank the author for their additions in the text. There are some minor editing requirements in some of the added text, so I suggest double checking those for editing requirements. (I have made two minor comments in the attached document).
I think the authors could highlight some of the confusions evident in the participants' perceptions and responses in their pointing to the need for ongoing training for HCPs concerning holistic care of queer individuals.

I thank the authors for having the manuscript edited. It is much improved.
Author Response
Thank you very much for taking the time to review our manuscript. We appreciate the positive feedback that will improve this manuscript. We have worked together as authors to ensure all the concerns, questions, advises, and recommendation made by you as reviewer 1 are addressed diligently. Please find the detailed responses below and the corresponding minor revisions highlighted submitted files.
Thank you, Reviewer 1, for the comments. We appreciate the comments and are all corrected. See the responses from us below:
Comment 1: Reviewer 1: This gives the impression that the participants themselves are queer, which, I take it, is not what you mean.
Response 1: This was an error. Queer word removed.
Comment 2: Reviewer 1: missing text
Response 2: Author 59 moved to the correct citing area.
Comment 3: Reviewer 1: editing required.
Response 3: Corrected and edited as “before commencement of data collection”
Reviewer 2 Report
Comments and Suggestions for Authors
The revised article brings new contributions to the quality of the work and will be used for new publications.
I agree with the answers given in the review.
Author Response
Thank you for agreeing with the answers given in the review.We appreciate your time for reviewing our manuscript for the second time.
No revision required.
Reviewer 4 Report
Comments and Suggestions for Authors Thank you for considering the suggestions.Author Response
Thank you for taking time to review our manuscript. No revision required.